# Pan-neuroblastoma analysis reveals age- and signature-associated driver alterations

Samuel W. Brady [1,9], Yanling Liu[1,9], Xiaotu Ma[1], Alexander M. Gout[1], Kohei Hagiwara[1], Xin Zhou[1], Jian Wang[1], Michael Macias [1], Xiaolong Chen [1], John Easton[1], Heather L. Mulder [1], Michael Rusch [1], Lu Wang [2], Joy Nakitandwe[2], Shaohua Lei[1], Eric M. Davis[1], Arlene Naranjo[3], Cheng Cheng[4], John M. Maris[5], James R. Downing[2], Nai-Kong V. Cheung [6], Michael D. Hogarty [7✉], Michael A. Dyer [8✉] & Jinghui Zhang [1✉]

Neuroblastoma is a pediatric malignancy with heterogeneous clinical outcomes. To better understand neuroblastoma pathogenesis, here we analyze whole-genome, whole-exome and/or transcriptome data from 702 neuroblastoma samples. Forty percent of samples harbor at least one recurrent driver gene alteration and most aberrations, including *MYCN*, *ATRX*, and *TERT* alterations, differ in frequency by age. *MYCN* alterations occur at median 2.3 years of age, *TERT* at 3.8 years, and *ATRX* at 5.6 years. COSMIC mutational signature 18, previously associated with reactive oxygen species, is the most common cause of driver point mutations in neuroblastoma, including most *ALK* and Ras-activating variants. Signature 18 appears early and is continuous throughout disease evolution. Signature 18 is enriched in neuroblastomas with *MYCN* amplification, 17q gain, and increased expression of mitochondrial ribosome and electron transport-associated genes. Recurrent *FGFR1* variants in six patients, and *ALK* N-terminal structural alterations in five samples, identify additional patients potentially amenable to precision therapy.

[1] Department of Computational Biology, St. Jude Children's Research Hospital, Memphis, TN, USA. [2] Department of Pathology, St. Jude Children's Research Hospital, Memphis, TN, USA. [3] Department of Biostatistics, University of Florida, Children's Oncology Group Statistics & Data Center, Gainesville, FL, USA. [4] Department of Biostatistics, St. Jude Children's Research Hospital, Memphis, TN, USA. [5] Division of Oncology and Center for Childhood Cancer Research, Children's Hospital of Philadelphia and the Perelman School of Medicine at the University of Pennsylvania, Philadelphia, PA, USA. [6] Department of Pediatrics, Memorial Sloan Kettering Cancer Center, New York, NY, USA. [7] Division of Oncology, Department of Pediatrics, Children's Hospital of Philadelphia, Perelman School of Medicine at the University of Pennsylvania, Philadelphia, PA, USA. [8] Department of Developmental Neurobiology, St. Jude Children's Research Hospital, Memphis, TN, USA. [9] These authors contributed equally: Samuel W. Brady, Yanling Liu. ✉email: hogartym@email.chop.edu; Michael.Dyer@StJude.org; Jinghui.Zhang@StJude.org

Neuroblastoma is among the most common childhood solid tumors, with diverse clinical behaviors ranging from spontaneous regression to progression despite aggressive therapy[1]. Neuroblastomas can be classified into low-, inter-mediate-, and high-risk groups based on clinical and molecular features[2]. Low-risk neuroblastomas mostly occur in children under 18 months of age, and standard care for these patients includes surgery alone or observation without surgery as many spontaneously regress[3]. Children in high-risk groups receive dose-intensive chemotherapy, surgery, radiation therapy, and immunotherapy over an 18-month timespan[4]. For children with intermediate-risk disease, surgery and outpatient chemotherapy can achieve a high survival rate[5]. Age plays a significant role in risk group classification and younger children overall have superior survival[2].

Genomic aberrations in neuroblastoma have been analyzed by multiple approaches. Whole chromosome gains are frequently observed in low-risk neuroblastoma[6], while gains or losses of chromosome arms (segmental chromosome alterations), includ-ing loss of 1p, 3p, 4p, and 11q and gain of 1q, 2p, and 17q, are associated with poor prognosis[7]. *MYCN* amplification is the most frequent driver in neuroblastoma, occurring in ~20% of cases and conferring poor prognosis[1]. The *SHANK2* tumor suppressor is frequently disrupted by structural variants[8], *TERT* activating alterations frequently occur in high-risk neuroblastoma[9], and *ATRX* inactivation occurs in a large proportion of adolescent and young adult neuroblastomas but rarely in those arising from younger patients[10]. Interestingly, *ATRX* alterations are mutually exclusive with *TERT*-activating variants[11] likely due to their overlapping functions in telomere maintenance[9,12]. Germline and somatic *ALK* kinase-domain point mutations have been detected in familial and sporadic neuroblastoma, respectively[13,14]. Addi-tionally, *ALK* N-terminal in-frame deletions or truncations have been reported in several neuroblastoma cell lines[15,16] and several primary neuroblastomas[16,17]. Functional studies have shown that these shortened *ALK* isoforms have oncogenic activity[15–18].

To better understand neuroblastoma pathogenesis, here we analyze whole-genome, whole-exome, and/or transcriptome sequencing data of 702 neuroblastomas comprised of all age and risk groups, assembled from the St. Jude/Washington University Pediatric Cancer Genome Project (PCGP), the Therapeutically Applicable Research to Generate Effective Treatment (TARGET) project, and 317 additional samples from the Children's Oncology Group (COG). Such a design allows identification of age-associated molecular aberrations in this developmental malig-nancy. The size of the cohort also allows identification of rare driver events.

## Results

**Landscape of somatic mutations in neuroblastoma.** We aggre-gated data from 702 neuroblastomas (679 diagnosis and 23 relapsed), which were profiled by whole genome sequencing (WGS, $n = 205$), whole exome sequencing (WES, $n = 539$), and/ or RNA-Seq ($n = 169$); 45% of the samples ($n = 317$) were new data generated from COG for this study (Supplementary Fig. 1a; the 317 samples were sequenced by WES plus targeted sequen-cing of the entire *ATRX* gene to detect structural variants along with sequence mutations). Of the 702 samples analyzed, 685 had DNA sequencing (WGS and/or WES, Supplementary Data 1; all but three had survival outcome information) while 17 had RNA-Seq alone. Single-nucleotide variants, indels, and segmental chromosome copy alterations were analyzed for all 685 samples with WGS or WES; structural variants and focal copy alterations were analyzed for the 205 samples with WGS; and *ATRX* was considered comprehensively analyzed in all 205 WGS cases as

well as 317 WES cases with *ATRX* targeted sequencing. The 23 relapse samples were excluded from survival analyses and age associations. The sample numbers and data sets used in each figure are described in figure legends. All disease stages were represented in the data set, with 60% of samples being stage 4 disease (Supplementary Table 1). The median age was 2.7 years old (interquartile range 1.2–4.6) and patients were categorized into three biologically and clinically relevant age groups[2,19,20]: group A (<1.5 years), group B (1.5–5 years), and group C (>5 years) with 206, 325, and 154 tumors, respectively (Supplemen-tary Fig. 1b). As expected, group A had significantly better out-comes than other groups, with 5-year survival of 89% compared to <60% in groups B and C ($P < 2 \times 10^{-16}$ by log-rank test, Supplementary Fig. 2).

To define the subgroup-specific genomic landscape, we identified somatic copy number variation (CNVs, Supplementary Data 2), structural variation (SVs, Supplementary Data 3), single-nucleotide variants (SNVs, Supplementary Data 4), and small insertion–deletions (indels; Methods section). Significantly enriched somatic alterations, as identified through MutSigCV[21], GRIN[22] (SNVs and indels), and GISTIC[23] (CNVs) software, are depicted in Fig. 1. Among segmental chromosome copy alterations, frequent copy number loss occurred at 1p, 3p, 4p, and 11q while copy number gain occurred predominantly on 1q, 2p, 7q, 11q13.3, 12q, and 17q (Fig. 1, Supplementary Figs. 3 and 4, and Supplementary Data 2). A total of 388 (57%) samples had at least one of these segmental copy alterations. Further, 31.4% of samples had nine or more whole-chromosome copy gains (9+ WC gains group, Fig. 1), a local minimum neatly separating ploidy by density plot analysis (Supplementary Fig. 5). The presence of nine or more whole-chromosome gains was more common in younger patients (group A, Fig. 1) and was associated with better survival, particularly in the absence of segmental chromosome alterations, consistent with previous findings[7] (Supplementary Fig. 6). We also found that copy number gains evolutionarily preceded the acquisition of most point mutations in 85% of samples, as most mutations in 3-copy regions affected 1 of 3 copies (occurring after the copy gain) and only a minority affected 2 of 3 copies (occurring before copy gains; Supplemen-tary Fig. 7), reinforcing the importance of copy number gains in early neuroblastoma pathogenesis.

Forty percent of samples had somatic alterations in known driver genes, and 62% of samples had driver gene alterations and/or recurrent segmental chromosome alterations shown in Fig. 1. The most frequently altered genes were *MYCN* (19% of samples; primarily amplification), *TERT* (17%; SVs), *SHANK2* (13%; SVs), *PTPRD* (11%; SVs and focal deletions), *ALK* (10%; SNVs and SVs), and *ATRX* (8%; multiple mutation types; Fig. 1). Among rare drivers altered in <5% of samples, we observed somatic mutations in *FGFR1* (1.0% of samples), *TP53* (0.9%), and the Ras pathway (4.2%). Mutations in genes involved in Ras signaling, including *PTPN11*, *NF1*, *NRAS*, *KRAS*, and *BRAF*, never co-occurred with one another although mutual exclusivity was not statistically significant due to low prevalence (Fig. 1). Each of the driver genes and segmental chromosome copy alterations shown in Fig. 1 have been identified as recurrent events in neuroblastoma[7–9,19,24–27], or in other cancer types (*FGFR1*[28]). Of the 685 samples with WGS or WES, 136 (20%) lacked any of these recurrent alterations and had no whole-chromosome copy alterations. These 136 samples were enriched in low disease stage (only 19% were stage 4 compared to 70% of other samples, $P < 2.2 \times 10^{-16}$ by Fisher's exact test) and their paucity of somatic alterations was not caused by low tumor purity—pathology review, available for 130 of 136, showed tumor purity exceeding 60% in all cases.

Alterations in *SHANK2*, a recently identified tumor suppressor in neuroblastoma[8] located on 11q13.3, most frequently occurred

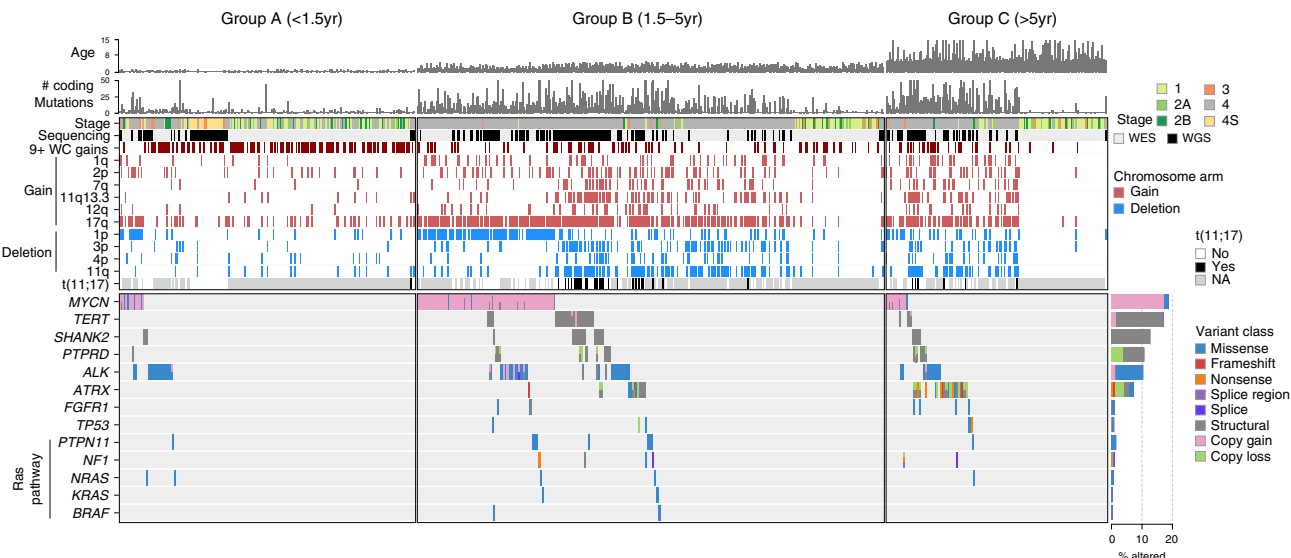

**Fig. 1 Recurrent somatic alterations by age group in neuroblastoma.** Top, age at diagnosis and number of coding mutations in each of 685 neuroblastoma samples (662 diagnosis, 23 relapse) sequenced by WGS or WES, with samples categorized into <18 months of age at diagnosis (group A, n = 206), 18 months to 5 years (group B, n = 325), and 5 or more years (group C, n = 154). Middle, segmental chromosome copy changes which were statistically significant per GISTIC analysis, and structural variants; blue indicates segmental copy loss, red indicates segmental copy gain, white indicates no change. 9+ WC gains samples (gain of nine or more whole chromosomes) are shown in dark red. Bottom, somatic variants in driver genes. Color indicates the type of mutation (key at right). Samples with more than one mutation type for a gene have multiple colors indicated. Significantly mutated genes (SMGs), as identified through MutSigCV and/or GRIN, range from *MYCN* at the top to *ATRX* at the bottom, while the remainder are known cancer genes that have pathogenic variants in our cohort but did not pass the SMG test. Barplot to the right shows percentage of samples with each gene somatically altered, and variant type indicated by color; the denominator was 685 samples for all alterations except for t(11;17), *TERT*, *SHANK2*, and *PTPRD* (205 samples with WGS as these variants required WGS to detect) and *ATRX* (522 samples including WGS samples plus WES samples with *ATRX* targeted sequencing allowing SV detection). Copy gains include focal *MYCN* amplifications with log$_2$ fold change of >2.0 (>8 copies); *ALK* copy gains meeting this criterion or one-copy *ALK* gains associated with a likely activating SV; or focal *TERT* gains of one copy or more.

by translocation t(11;17) or other types of SVs disrupting the gene (Fig. 1, Supplementary Fig. 8a, and Supplementary Data 3). Interestingly, *SHANK2* disruption and t(11;17) were frequently accompanied by 11q13.3 copy number gains associated with the chromosome 11 breakpoint, where six genes at this locus showed increased expression in 11q13.3-gained samples, including *CCND1*, a cancer gene with well-established oncogenic activity in neuroblastoma[29] (Supplementary Fig. 8a, bottom, P = 0.003 by two-sided Wilcoxon rank-sum test). The *CCND1* expression increase in 11q13.3-gained samples was not confounded by stage as the association remained significant when including only stage 4 samples (Supplementary Fig. 8b, P = 0.005 by two-sided Wilcoxon rank-sum test). *SHANK2* and t(11;17) alterations were also associated with chromosome 17q gains and increased expression of 17q genes, and t(11;17) frequently joined the termini of 17q and 11q copy alterations (Supplementary Fig. 8a). Twenty of 35 t(11;17) events did not directly disrupt *SHANK2*, and t(11;17) translocations' association with 17q gains, 11q13.3 gains, and *CCND1* copy and expression increases suggests that *SHANK2* is not the only target gene affected by t(11;17) (Supplementary Fig. 8a).

*MYCN*, *ALK*, *ATRX*, and Ras pathway alterations; segmental deletion of 1p, 3p, and 11q; and segmental gain of 1q, 2p, 7q, 11q13.3, 12q, and 17q were each significantly associated with poor overall survival (Supplementary Fig. 9), as reported previously[1,7,30,31]. *MYCN* and *ALK* co-occurrence predicted particularly poor survival (Supplementary Fig. 10), confirming findings from animal models[32]. *FGFR1* and *TERT* alterations trended towards worse survival but were not statistically significant (P = 0.09 and P = 0.1, respectively, by log-rank test; Supplementary Fig. 9). In previous studies, *TERT* has been associated with both worse survival[9] and no difference in

survival[33]. These reported findings, together with ours, suggest that *TERT* alterations alone have a modest effect on patient outcomes.

**Kinase alterations in FGFR1 and truncated ALK variants.** We observed seven somatic mutations in *FGFR1*, including N546K mutations in six samples from five patients (at both diagnosis and relapse in one patient) and an internal tandem duplication (ITD) in the kinase domain of one additional sample (Fig. 2a), out of the 685 samples with WGS or WES. *FGFR1* N546K was previously reported in a single neuroblastoma patient[34] and therefore has not been considered a driver gene in neuroblastoma. This variant activates MAPK signaling in another tumor type[28], and is recurrent in pediatric low-grade glioma[35], indicating it is a driver mutation. All *FGFR1* N546K variants were clonal (variant allele fraction (VAF) >0.4 when diploid; see Methods section) while the ITD was subclonal, and these variants co-occurred with *MYCN* or *ATRX* alterations in five out of the six patients (P = 0.004 by Fisher's exact test). Further, the median *FGFR1* expression was ranked at the 61st percentile of expression in this cohort (median transcripts per million (TPM) of 16.3), indicating that *FGFR1* is expressed in neuroblastoma. Specifically, in the two *FGFR1*-mutant samples that also had RNA-Seq, *FGFR1* expression was at 79.6 TPM (the 87th percentile of expression within the sample) or 21.6 TPM (69th percentile), and the mutant *FGFR1* alleles were expressed at similar allele frequencies as in DNA (0.30 VAF in RNA vs. 0.22 in WGS, in one example patient).

*ALK* was altered in 10% of samples, including known hotspot SNVs at F1174, F1245, and R1275[18] (n = 61), amplification to 10 or more copies (n = 9), and structural alterations (n = 5) which removed some or all of exons 1–4 while preserving the transmembrane and kinase[36] domains (Fig. 2b). The five

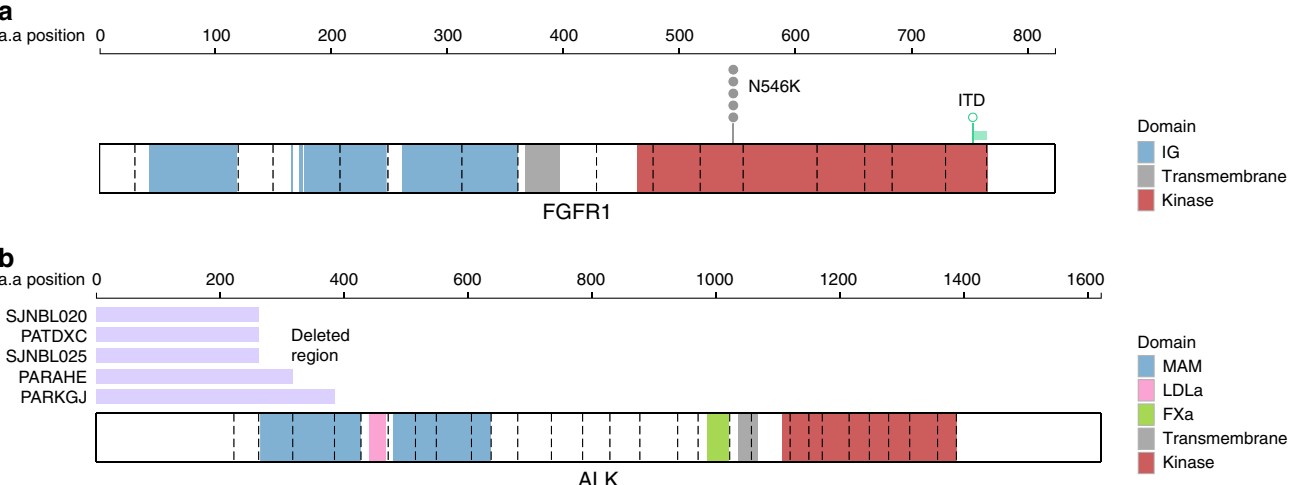

**Fig. 2 FGFR1 kinase alterations and ALK N-terminal structural alterations. a** Positions of point mutations in FGFR1 in six patients, including 5 SNVs at N546K and an ITD in the kinase domain (duplicated region in green). IG indicates the extracellular immunoglobulin-like domain (blue), while the intracellular kinase domain is shown in red. Exon boundaries are indicated by dotted lines. **b** ALK protein structure with protein domains indicated by color. Dotted lines indicate exon boundaries. The five samples with ALK N-terminal structural alterations are shown, with the lost exons indicated in purple. MAM indicates Meprin A5 and tyrosine phosphatase Mu domain; LDLa indicates low-density lipoprotein receptor domain class A domain; FXa indicates coagulation factor Xa inhibitory site domain.

structural events occurred through exon 1–4 deletion in one sample (Supplementary Fig. 11a), tandem duplications removing exons 1–2 or 1–3 in two samples (Supplementary Fig. 11b, c), or inter-chromosomal translocations in intron 2 resulting in removal of exons 1–2 in two samples (Supplementary Fig. 12, Fig. 2b, and Supplementary Data 5), and were similar to ALK truncations previously reported in neuroblastoma[15–18]. One of these two translocations occurred in sample SJNBL020, and was a t(2;8) event bringing truncated ALK (exon 3 to final exon 29) into proximity to MYC on chromosome 8, including several enhancers near MYC which are active in neuroblastoma[37], though any effect on ALK transcription could not be assessed due to lack of RNA-Seq (Supplementary Fig. 12a). The other translocation, in sample PATDXC, was a t(2;5) bringing truncated ALK (exons 3–29) adjacent to the TERT promoter. TERT expression was in the 79th percentile, and mono-allelic expression was detected in a run of seven heterozygous SNPs in or around TERT, consistent with cis-activation of a single TERT allele due to the nearby translocation (Supplementary Fig. 12b). The truncated ALK isoforms in these five samples may constitutively activate ALK signaling, as N-terminal deletions have also been reported in several neuroblastoma cell lines and patient samples[15–18]. Indeed, deletion of exons 2–3 activates ALK[15], indicating abrogation of an inhibitory function of this domain. In addition to the five cases referred to above, we observed alternative splicing of ALK in three samples which would lead to loss of N-terminal exons (Supplementary Fig. 13), although WGS data were not available for assessing causality by genetic alterations.

**Mutual exclusivity and co-occurrence of alterations**. We analyzed the significance of mutual exclusivity and co-occurrence of recurrent somatic variants using WGS diagnosis samples (n = 182) due to their comprehensively characterized variant sets, including SVs (Fig. 3a), and events with low prevalence (<5 events, such as FGFR1 which had only two mutations among WGS samples) were not included due to lack of power. Our analysis focuses on variants that were independent which excludes CNVs and SVs resulting from the same re-arrangement event (Fig. 3a). MYCN alterations were mutually exclusive with ATRX and SHANK2; with t(11;17); with deletion of 3p, 4p, and

11q; and with gain of 11q13.3; each of these associations has been reported previously[6,8,10,26,38,39]. TERT was mutually exclusive with ATRX as previously reported, as both genes promote telomere lengthening[9,12]. MYCN and TERT were not mutually exclusive as three cases harbored independent MYCN and TERT alterations (Fig. 3a), which agrees with one previous study[9] but contradicts another[40]. Interestingly, two additional cases harbored SVs simultaneously affecting MYCN and TERT (Supplementary Fig. 14). ALK mutations did not exhibit statistically significant co-occurrence with any gene-specific events (Fig. 3a). Deletion of 3p, 4p, and 11q, and gain of 7q, tended to co-occur, while 1p deletion was mutually exclusive or not co-occurrent with these alterations, consistent with previous findings[39] (Fig. 3a and Supplementary Fig. 4). Gain of 17q co-occurred with most other segmental chromosome alterations, as expected[7] (Fig. 3a and Supplementary Fig. 4).

**Age-related genomic aberrations**. Mutual exclusivity in driver alterations prompted us to analyze whether it was caused by differential prevalence of these events among the three age groups. Globally, group A showed the lowest mutation burden compared with groups B and C based on SNV count in coding regions (Fig. 3b). Alterations affecting ALK (by SNVs and SVs) and SHANK2 (by SVs) did not exhibit significant bias among age groups (Supplementary Fig. 15b). While SVs disrupting SHANK2 were present in all age groups, inter-chromosomal translocations occurred in the older age groups (groups B and C) but not in group A samples which were affected by intra-chromosomal disruptions with breakpoints ~2.5 Mb apart. By contrast, PTPRD genetic alterations (consisting of gene-disrupting SVs and focal deletions) were significantly higher in groups B and C as they were completely absent from group A (Supplementary Fig. 15a, b), and Ras pathway mutations were enriched in group B, although there was no significant age group difference for PTPRD and Ras pathway mutations when including only stage 4 samples, suggesting their age specificity was related to higher disease stage (Supplementary Fig. 15b). MYCN and TERT alterations were enriched in younger patients (median age of 2.3 years and 3.8 years, respectively) in group B, while ATRX was more common in older children (median age 5.6 years) in group C ($P < 1 \times 10^{-3}$ for

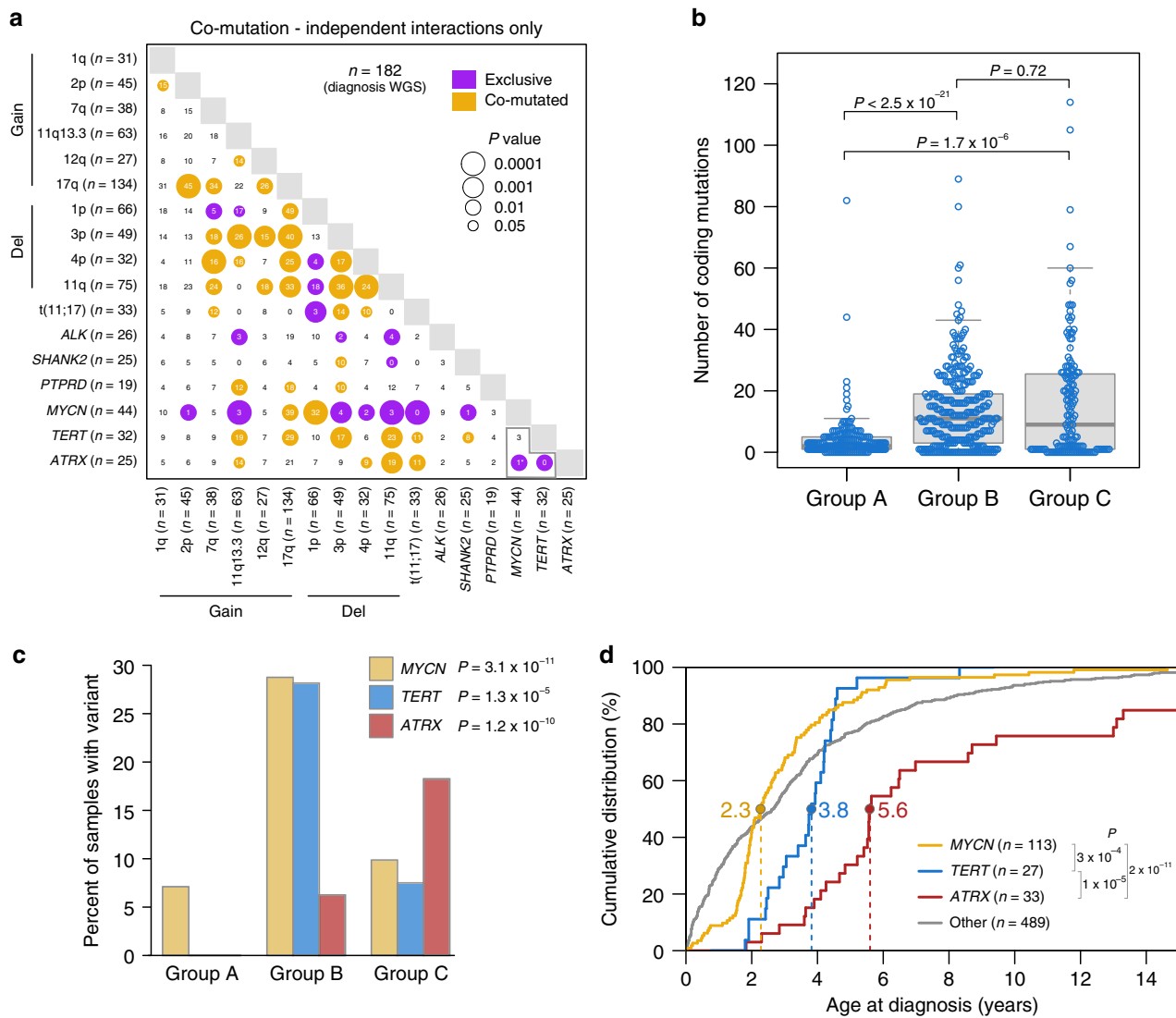

**Fig. 3 Age-associated genetic alterations. a** Mutual exclusivity and co-occurrence in diagnosis samples with WGS (n = 182). When two alterations co-occurred in a sample, the alterations were excluded from analysis if the two alterations were joined by an SV (non-independent). P values are by two-sided Fisher's exact test. Asterisk, female with ATRX and MYCN alterations. Gold, significant co-occurrence; purple, mutually exclusivity. Circle size is inversely related to P value. Gains and deletions (del) include segmental chromosome alterations and exclude whole-chromosome alterations. **b** Boxplot showing coding mutation burden, including 662 diagnosis samples with WGS or WES; a single outlier in group B had 200 mutations (sample numbers: group A, 197; B, 313; and C, 152). Box, interquartile range (25th to 75th percentile); middle bar, median. Whiskers are described in R boxplot documentation (a 1.5 × interquartile range rule is used). Each point represents one sample and P values are by two-sided Wilcoxon rank-sum test. **c** Percentage of diagnosis samples with indicated gene alterations by age group; n = 662, 182, or 499 for MYCN (WGS or WES samples), TERT (WGS samples), or ATRX (WGS samples or WES samples with ATRX targeted sequencing), respectively. P values are by two-sided Fisher's exact test comparing prevalence among age groups. **d** Empirical cumulative distribution function showing diagnosis age (x-axis) of patients with alterations in MYCN (gold), TERT (blue), ATRX (red), or none of these alterations (gray, other). Five patients with both MYCN and TERT alterations and one with both MYCN and ATRX were included in the other group. 662 diagnosis samples with WGS or WES were analyzed. Y-axis represents the percent of patients in each group diagnosed at or before the age indicated on x-axis. P values are by two-sided Wilcoxon rank-sum test. Dotted lines indicate median age in the mutated group. Several ATRX-mutant patients were diagnosed at >15 years; hence the ATRX curve does not reach 100%. If only WGS data were considered, the MYCN median age was 2.5, the TERT and ATRX median ages did not change, and the P value comparisons shown were P < 0.003. Source data are provided as a Source Data file.

each pairwise comparison by Wilcoxon rank-sum test; Fig. 3c, d; see also Supplementary Fig. 15). These correlations remained significant even when including only stage 4 samples (Supplementary Fig. 15b). Thus, different developmental stages may be more vulnerable to the acquisition of specific oncogenic mutations.

Among segmental chromosome alterations, 1p deletion showed the highest frequency in group B, while frequently co-occurring 3p deletions, 11q deletions, and 7q gains were enriched in groups B and C, whether analyzing all samples or stage

4 samples only (Supplementary Fig. 15b). t(11;17) events likewise showed significantly higher prevalence in groups B and C, though there was no significant correlation when analyzing only stage 4 samples (Supplementary Fig. 15b).

**Mutational signatures and genetic correlates of signature 18.** To identify the processes driving mutagenesis, we analyzed SNV mutational signatures based on each mutation's flanking trinucleotide context[41]. We extracted mutational signatures from the

205 WGS samples, since WES data generally lack sufficient SNVs to identify reliable signatures. We identified five known mutational signatures in primary neuroblastomas, consistent with our previous report[11] (Fig. 4a, showing diagnosis samples only); one relapse-specific signature induced by cisplatin (Supplementary Fig. 16, showing diagnosis and relapse); and one signature representing artefacts from Complete Genomics Inc. (CGI) sequencing[11] (T-10; Fig. 4a and Supplementary Fig. 16). Most of the signatures were present across all age groups, and represent the following biological processes in order of abundance in neuroblastoma: (1) signature 18, which has been causally associated with reactive oxygen species (ROS)-induced mutagenesis in cultured iPS cells and mouse models[42,43]; (2) signature 3, putatively caused by homologous recombination deficiency; (3 and 4) clock-like signatures 5 and 1, indicative of unclear causes and 5-methylcytosine deamination, respectively; (5) signature 31, which is cisplatin-induced and was found exclusively at relapse in 5 of 23 relapses[41,44]; and (6) signature 40, of unknown etiology and found in a single sample (Fig. 4a and Supplementary Fig. 16).

Signature 18 has predominantly been detected in neuroblastoma but is rare in most other cancers[41] and is induced by ROS in some model systems[42,43]. It primarily induces $C > A$ transversions[41] potentially through ROS-induced 8-oxoguanine formation[42,45], though whether signature 18 is caused by ROS in neuroblastoma specifically is unknown. We focused analysis on signature 18 due to its specificity for neuroblastoma compared to ubiquitous signatures (signatures 1 and 5), and its unique preference for $C > A$ mutations, enabling identification of driver mutations potentially caused by signature 18, compared to less-discriminatory signatures causing various mutation types with similar frequency (signatures 3 and 5)[41,46].

We found that signature 18 may be both an early event in neuroblastoma, causing truncal mutations, and an on-going mutational event causing relapse-specific mutations, based on analysis of five patients with matched diagnosis and relapse samples (Fig. 4b), as suggested previously by cultured neuroblastoma cell models[47]. Four of the five patients had signature 18 in both shared (early) and relapse-specific (late) mutations, while the fifth lacked the signature at both diagnosis and relapse (Fig. 4b). This suggests an intrinsic and stable propensity of specific neuroblastoma tumors to either possess or lack the signature 18 mutational process.

The prevalence of signature 18 varied considerably, ranging from 0 to 3302 SNVs per tumor (Supplementary Fig. 16). To identify biological processes that may contribute to this variability, we compared gene expression profiles between samples with detectable signature 18 vs. those lacking signature 18 altogether. This analysis included 88 diagnosis samples that had both WGS and RNA-Seq, 28 of which were signature 18-negative and 60 of which were signature 18-positive. Differential expression was analyzed using Limma to generate Benjamini–Hochberg-adjusted $P$ values for each gene (Fig. 4c). Genes involved in neural function, including *PIRT* (affecting peripheral nerve function[48]), *TMEFF2* (which promotes neuron survival[49]), *DST* (a neural cytoskeletal gene[50]), and *GABBR1* (a neurotransmitter receptor[51]), were among the most significantly increased in expression in signature 18-negative samples. On the other hand, genes statistically increased in signature 18-positive samples included *UBE2S* and *LRRC59*; nuclear-encoded mitochondrial ribosome genes *MRPL11*, *MRPS7*, and *ICT1* (also called *MRPL58*); other nuclear-encoded mitochondrial genes *COX8A* (a subunit of electron transport chain complex IV[52]), *ATP5G1* (ATP synthase subunit[52]), *MTFP1* (promotes mitochondrial fission[53]); and the histone genes *HIST1H4H*, *HIST1H2AG*, and *HIST1H3H* (Fig. 4c, d). Indeed, 16.4% of all significantly up-regulated genes in signature 18-positive samples (genes marked

red in Fig. 4c) had mitochondrial localization according to the Broad Institute's MitoCarta 2.0 database[54], compared to only 6.8% of all other genes ($P = 0.016$ by Fisher's exact test; see accompanying Source Data file). Interestingly, several of the mitochondrial genes were found on chromosome 17q, including *ATP5G1*, *ICT1*, and *MRPS7*.

Indeed, 17q gain itself was significantly associated with increased signature 18 proportion (Fig. 4e and Supplementary Fig. 17; $P = 1.3 \times 10^{-9}$ by Wilcoxon rank-sum test). Further, the neural-enriched samples, with neural expression scores determined using ssGSEA[55] of 26 neural genes statistically increased in signature 18-negative samples (Fig. 4c and Supplementary Table 2), clustered together in t-SNE analysis as a group lacking 17q gain (Fig. 4f, left two panels) and signature 18 (Fig. 4f, right panel). These data suggest that the neural-enriched samples lacking signature 18 may represent a distinct subgroup; indeed, a neuronally differentiated transcriptional subgroup was previously identified in neuroblastoma[56]. We also tested whether other genetic alterations, in addition to 17q gain, were associated with signature 18 (Supplementary Fig. 17). *MYCN*-altered samples had significantly more signature 18 (Fig. 4e), consistent with reported *MYCN*-induced ROS generation in neuroblastoma[57], as did samples with 1p deletion and 2p gain ($P < 3 \times 10^{-4}$ for each of these alterations by Wilcoxon rank-sum test; we required a significance level of $\alpha = 2.94 \times 10^{-3}$ using a Bonferroni adjustment for multiple hypothesis testing of the 17 alterations in Supplementary Fig. 17). *MYCN* and 17q gains were also statistically enriched in signature 18-positive samples when including only stage 4 samples (Supplementary Fig. 18) and the structural variant burden was not strongly correlated with signature 18 (Pearson $r^2 = 0.05$), indicating that *MYCN* and 17q gains' association with signature 18 was not simply due to higher disease stage or genome complexity.

Samples with 17q gain but no *MYCN* alteration had statistically higher signature 18 than samples lacking either alteration, indicating that 17q gain's association with signature 18 was not simply due to its co-occurrence with *MYCN* (Supplementary Fig. 19a). Further, samples with both 17q gain and *MYCN* alterations had even higher signature 18 than those with 17q gain alone, suggesting additive interaction (Supplementary Fig. 19a). Finally, *MYCN* was not the primary driver of increased mitochondrial gene expression in signature 18-positive samples, since mitochondrial genes remained statistically increased when differential gene expression analysis included *MYCN* alteration status as a covariate, to remove potential confounding effects of *MYCN* (Supplementary Fig. 19b). This is consistent with 17q gains potentially promoting the mitochondrial gene expression, given that several of the mitochondrial genes are found on 17q. Indeed, when both 17q gains and *MYCN* status were included as covariates, signature 18 alone was not significantly associated with expression of any gene, and mitochondrial gene expression was associated with 17q gains (Supplementary Fig. 19c) but not *MYCN* alterations (Supplementary Fig. 19d). This indicates that 17q gains were the primary factor associating signature 18 with mitochondrial gene expression. 17q genes with mitochondrial localization per MitoCarta[54] had a statistically greater expression increase in 17q-gain samples (median adjusted $P = 0.006$ by Limma analysis), compared to non-mitochondrial 17q genes (median adjusted $P = 0.038$ by Limma analysis; comparison of the adjusted $P$ values between the two groups by a Wilcoxon rank-sum test yielded $P = 0.042$).

To test whether similar signature 18 correlations could be observed in other cancer types, we analyzed mutational signatures in 1603 pediatric cancers with WGS spanning 39 cancer types including all major pediatric cancers, and 831 adult cancers from TCGA[58] with WGS spanning 24 cancer types. Pediatric

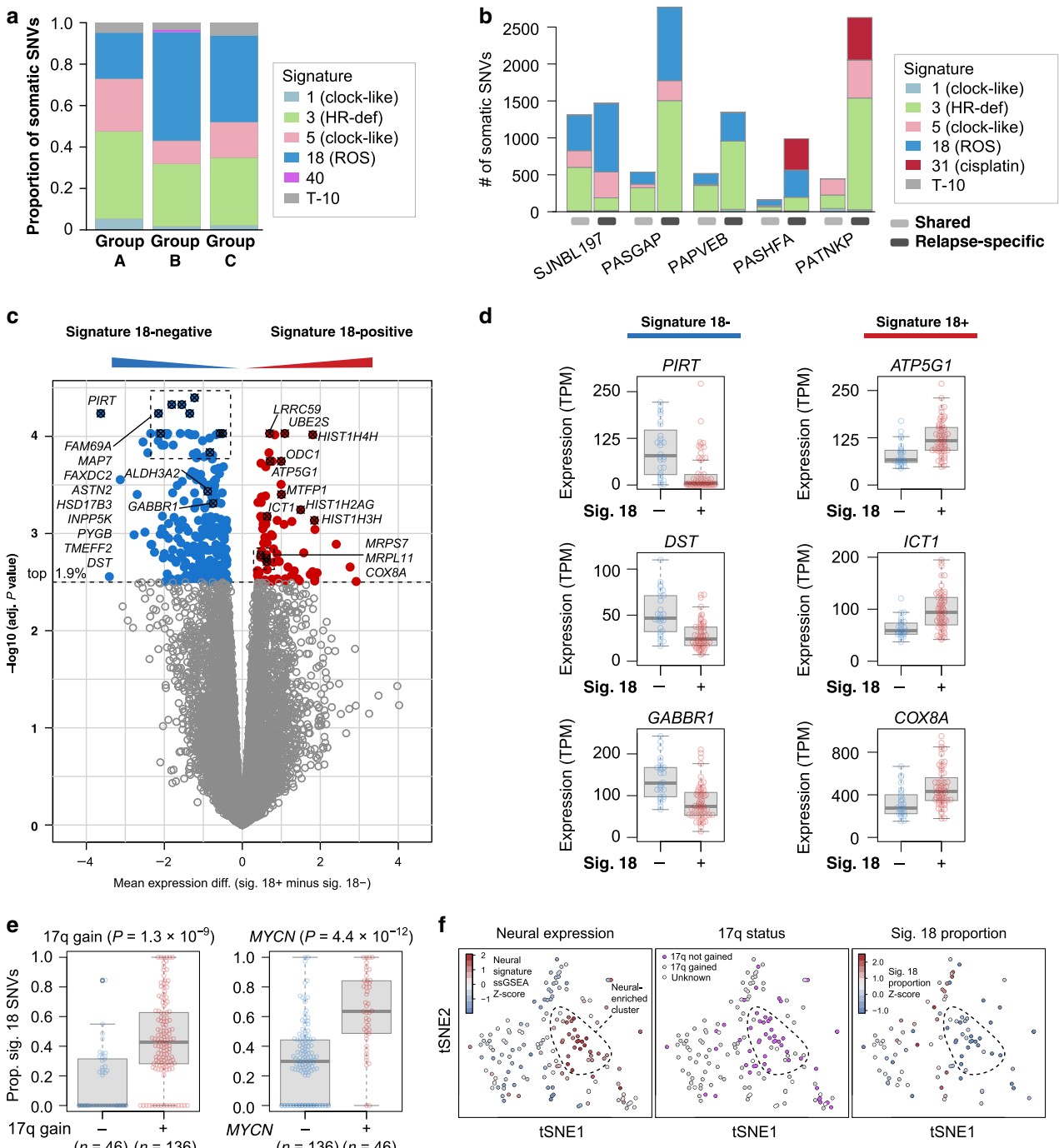

**Fig. 4 Signature 18-induced mutational patterns. a** Mutational signatures detected by WGS of 182 diagnosis samples, shown as proportion of mutations caused by the signature across all samples in each age group. Sample numbers are: group A, 39; B, 103; and C, 40. **b** Signature abundance in absolute SNVs is indicated on *y*-axis for five patients with matched diagnosis and relapse samples. Shared mutations (present in both diagnosis and relapse, which are thus early mutations) are indicated at left, followed by relapse-specific variants (detected only at relapse) on the right for each patient. **c** Differential gene expression analysis comparing signature 18-positive (*n* = 60) vs. signature 18-negative (*n* = 28) samples in 88 diagnosis samples with both WGS and RNA-Seq using Limma. *Y*-axis represents -$\log_{10}$ *P* values; *x*-axis represents the log expression fold-change (mean expression difference), with genes increased in signature 18-positive samples in the positive direction (red) and genes increased in signature 18-negative samples in the negative direction (blue). Each point represents one gene, and genes with the lowest 1.9% of adjusted *P* values are shown in color. **d** *Y*-axis represents indicated gene's expression in TPM, comparing signature 18-negative (*n* = 28) and -positive (*n* = 60) samples. Box, interquartile range (25th to 75th percentile); middle bar, median. Whiskers are described in R boxplot documentation (a 1.5 × interquartile range rule is used). **e** Proportion of SNVs caused by signature 18 in 182 diagnosis samples with WGS, comparing samples with vs. without 17q gain; or with vs. without *MYCN* alterations. Interquartile range, median, and whiskers are as in **d**. *P* values are by two-sided Wilcoxon rank-sum test. **f** t-SNE of 158 diagnosis and relapse samples with RNA-Seq (left two panels) or the subset of these with WGS (*n* = 96, far right). Overlaid are, from left to right: (1) ssGSEA *Z*-scores for the neural gene set (Supplementary Table 2) with red indicating more neural expression, (2) the 17q gain status with purple indicating no gain, and (3) proportion of mutations caused by signature 18 in each sample (*Z*-scores). Dotted outline is to enable comparison of a neural-enriched group between each plot. Source data are provided as a Source Data file.

rhabdomyosarcoma and liver malignancies, as well as adult colorectal cancers, were the only other solid tumor types with >20% of tumors bearing signature 18. As only three signature 18-positive liver tumors were available, and the colon epithelium is subjected to continuous environmental exposures which may confound the search for endogenous signature causes, we focused analysis on rhabdomyosarcoma which had 13 signature 18-positive samples out of 31 samples total, sequenced by both WGS and RNA-Seq. We performed differential gene expression analysis with Limma, comparing signature 18-positive vs. signature 18-negative samples (Supplementary Fig. 20a, b). This revealed statistically increased expression (adjusted $P$ value < 0.05) of four nuclear-encoded mitochondrial ribosome genes[59] (*MRPL15*, *MRPL47*, *MRPL33*, and *MRPL13*) and several nuclear-encoded components of the mitochondrial electron transport chain, including complex I components (*NDUFB3* and *NDUFB9*), one gene in complex II (*SDHD*), one gene in complex III (*UQCRB*), cytochrome c (*CYCS*) itself (which transports electrons between complexes III and IV), three genes in complex IV (*COX6C*, *COX6A1*, and *COX8A*), and two subunits of ATP synthase (including *ATP5MD* and *ATP5MPL*)[52]. The specific nuclear-encoded mitochondrial genes in rhabdomyosarcoma were different from those enriched in neuroblastoma signature 18-positive samples, except for *COX8A* (Fig. 4c, d). Several of the up-regulated mitochondrial genes were found on chromosome 8 (*COX6C*, *NDUFB9*, *UQCRB*, *MRPL15*, and *MRPL13*), which is commonly affected by copy gain in rhabdomyosarcoma[60] (Supplementary Fig. 20b). Indeed, rhabdomyosarcomas with chromosome 8 gain had significantly more signature 18 proportions ($P = 0.0036$ by Wilcoxon rank-sum test; Supplementary Fig. 20c), suggesting that the increased mitochondrial gene expression may have been driven by chromosome 8 gains. Together, these results suggest that certain copy gains (e.g. 17q gains in neuroblastoma, chromosome 8 gains in rhabdomyosarcoma) may lead to increased activity of mitochondria where ROS are frequently generated[61], potentially leading to signature 18[42,43]. However, experimental data are ultimately needed to test this hypothesis.

**Driver mutations potentially induced by signature 18.** We next analyzed which driver SNVs were likely induced by signature 18. Signature 18 is dominated by C>A mutations, and many driver SNVs were indeed C>A variants at signature 18-associated trinucleotide contexts, including *ALK* F1174L, R1275L, and L1196M; *NRAS* Q61K and *KRAS* G12V; *ATRX* 1690D and E990*; *NF1* E1868* and E977*; *PTPN11* D61Y; and *FGFR1* N546K (Fig. 5a), suggesting that signature 18 may induce driver mutations. We quantified the probability that driver mutations in these genes (along with *BRAF*, *TP53*, and *MYCN*) were induced by each mutational signature using an approach described previously[62] which we have applied in other pediatric cancers[63] (Fig. 5b; an example of how the probability calculations were performed for a specific example case is shown in Supplementary Fig. 21). Among the 161 WGS samples with signature reconstruction cosine similarity of 0.9 or above (as the association of individual mutations to signatures requires stringently accurate signature scores), 38 diagnosis samples had at least one SNV in one of the driver genes (e.g. *ALK*, *ATRX*, *TP53*, *FGFR1*, *MYCN*, *NF1*, *KRAS*, *NRAS*, *PTPN11*, and *BRAF*), for a total of 42 driver SNVs. The majority (22 of 42, or 52%) of these driver SNVs had >50% probability of having been induced by signature 18 (Fig. 5b), including five *ALK* F1174L mutations, the most common driver SNV, at the T[C>A]A context, and other *ALK* mutations at R1275 and L1196M. *ALK* F1245V (C[T>G]T), by contrast, was likely caused by signature 3 or 5. Eight of nine mutations affecting

the Ras pathway (in *NF1*, *NRAS*, *KRAS*, *BRAF*, or *PTPN11*) were most likely (>50% probability) induced by signature 18. Among mutations leading to genome instability (in *TP53* or *ATRX*), 3 of 11 were most likely induced by signature 18. The percent of driver SNVs most likely induced by signature 18 (52%) was similar to the percentage of all mutations caused by signature 18 across the 38 samples analyzed (56%, $P = 0.64$ by Fisher's exact test), indicating that signature 18 was proportionally likely to cause driver SNVs as SNVs in general. This indicates that signature 18 is likely a driver of disease progression in neuroblastoma, in contrast with passenger mutational signatures, such as the kataegis-associated APOBEC signature in osteosarcoma which causes no known driver SNVs in that cancer type[64].

**Discussion**

We previously reported age-associated *ATRX* alterations in a cohort of 104 neuroblastoma patients[10]. The comprehensive genome-wide analysis performed here allowed us to discover age-associated alterations in *MYCN*, *TERT*, *PTPRD*, and Ras pathway alterations, which together with *ATRX* represent the majority of common driver gene alterations in neuroblastoma. These findings suggest that the sympathetic nervous system, the tissue from which neuroblastoma arises, is susceptible to different oncogenic insults at different times during development, which could be explored in future investigations using animal models. Notably, telomere lengthening can be achieved either by telomerase expression through SVs in *TERT* or *MYCN* amplification (which is associated with higher *TERT* expression[9,65]), or by an alternative lengthening mechanism via *ATRX* mutation[12,57]. We observed age-associated mutual exclusivity between *ATRX* and *TERT*, and between *ATRX* and *MYCN*, indicating susceptibility of different ages to specific oncogenic events. This may be due to (1) age-specific oncogene dependencies in the tissue of origin, or (2) different proliferation rates conferred by specific alterations, leading to clinically detectable disease at earlier or later ages.

*ALK* N-terminal variants, present in 5 of 205 (2.4%) WGS samples, removed extracellular domain-encoding exons and were similar to aberrations observed previously[16,17]. Alterations removing ALK N-terminal exons activate ALK[15], suggesting they are possible therapeutic targets. While attempts to inhibit ALK in neuroblastoma have been disappointing[66], these efforts have focused on patients with kinase domain point mutations. Since the kinase domain is preserved in the ALK N-terminal variants, functional and pharmacological studies are merited to determine whether ALK kinase inhibitors are effective against these alterations. In addition, we found recurrent *FGFR1* N546K mutations in six patients or ~1% of total samples, in addition to the single neuroblastoma case reported previously[34]. Functional and pharmacological studies would likewise be beneficial to test whether *FGFR1* N546K and the *FGFR1* ITD we observed in one additional patient represent valid targets for kinase inhibition in neuroblastoma patients.

Our findings indicate that the signature 18 mutational process is the most common cause of driver SNVs in neuroblastoma. This suggests that this mutagenic process, which is caused by ROS in other settings[42,43] (though not proven in neuroblastoma), may promote evolution and heterogeneity, as many driver SNVs, such as *ALK* mutations, are later events in neuroblastoma[11,34]. Mitochondrial ribosomal and electron transport chain gene expression was associated with higher signature 18 in neuroblastoma—a finding that we were able to replicate in rhabdomyosarcoma (Fig. 4c, d and Supplementary Fig. 20a, b)—potentially through 17q gains, where several of these genes reside. Indeed, mitochondrial electron transport chain components I and III are known to generate ROS[61], a reported cause of signature 18 in

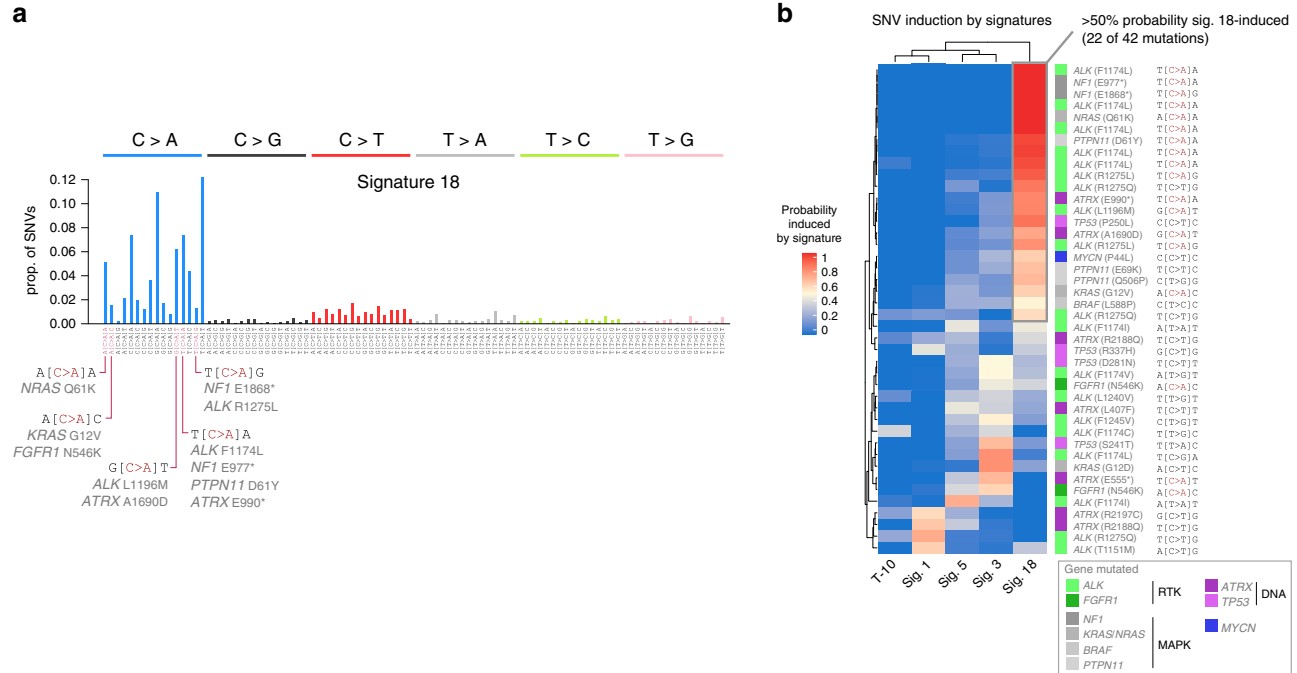

**Fig. 5 Driver SNVs caused by signature 18. a** Spectrum of signature 18, with 6 SNV types indicated at top and the trinucleotide context indicated at bottom. C>A mutations commonly affecting driver genes are indicated at bottom. **b** Heatmap showing probability that each somatic driver SNV (rows) was caused by each signature (columns). Each row represents one mutation in a specific patient. Trinucleotide context of each mutation is indicated at right. Genes are color-coded as indicated in legend at bottom-right. Only driver SNVs in diagnosis WGS samples with highly reliable signature data (cosine reconstruction similarity of 0.9 or higher; $n = 38$ samples and 42 mutations) were analyzed. Source data are provided as a Source Data file.

certain experimental models[42,43], and increased mitochondrial ribosome proteins may promote increased translation of complex I, some of whose subunits are translated by mitochondrial ribosomes[52]. We speculate that the poor outcomes in neuroblastomas with 17q gain may be partly due to their increased mutational burden (Fig. 1), which may result from 17q copy gains leading to increased expression of electron transport chain genes on 17q (or 17q mitochondrial ribosome genes that increase translation of electron transport chain components). This in turn may cause ROS production and potentially signature 18-induced mutagenesis, which fuels point mutations and clonal heterogeneity promoting drug resistance. Experimental models are ultimately needed to test these hypotheses. Indeed, mitochondrial metabolism has also been suggested to play a role in neuroblastoma pathogenesis by others[67]. Together, our findings suggest that there may be therapeutic vulnerabilities in neuroblastomas with 17q gain, potentially through targeting altered mitochondrial function.

## Methods
**Patients and samples**. Neuroblastoma specimens were collected through collaboration with Memorial Sloan-Kettering Cancer Center (MSKCC) and the Therapeutically Applicable Research to Generate Effective Treatments (TARGET) project and the Children's Oncology Group (COG). Neuroblastoma tissue was obtained from COG member institutions and was used under institutional review board (IRB) approval from The Children's Hospital of Philadelphia, MSKCC, and St. Jude Children's Research Hospital. Rhabdomyosarcoma PCGP and all St. Jude Clinical Genomics samples were obtained from St. Jude under IRB approval from that institution. Written informed consent was obtained from patients and/or legal guardians for use of tissue for research. This study complies with the Declaration of Helsinki and all other relevant ethical regulations.

**Statistics and reproducibility**. All statistical analyses were performed using two-sided tests if applicable. Specific tests used are described in the main text or figure legends. When comparing distributions we used the non-parametric Wilcoxon rank-sum test due to frequently skewed distributions. Each sample was sequenced one time and no technical replicates were performed.

**Genomic datasets**. Neuroblastoma WGS data in the PCGP cohort (samples from MSKCC) were reported previously[10]. WGS, WES, and RNA-seq data from TARGET were downloaded from dbGaP with study identifier phs000218. WES data for 317 COG samples were generated at St. Jude Children's Research Hospital. TARGET samples were sequenced by CGI technology for WGS, while the remainder of samples relied on Illumina sequencing processed with standard Illumina instrumentation software (primarily HiSeq2000). All somatic alterations identified in our study can also be explored interactively in our pediatric cancer data portal (https://pecan.stjude.cloud/proteinpaint/study/PanNeuroblastoma.Alterations). Rhabdomyosarcoma samples were from the PCGP and St. Jude Clinical Genomics programs.

**Copy number variation analysis**. CONSERTING version 1.0 was used to identify CNVs[68] after BWA (version 0.5.9) alignment in samples with Illumina-based WGS. For TARGET CGI-based WGS data, we used CNV data from our previous study, in which we adapted the CONSERTING algorithm to call CNVs[11]. After CNVs were called, significant CNV regions were identified with GISTIC version 2.0[23] using WGS data from 205 samples. Focal CNVs with targeted genes were manually curated afterwards (Supplementary Data 6). For WES data, CNVs were called with CnvKit[69] (version 0.9.1) after alignment with BWA version 0.5.9. CNV data were manually inspected by plotting copy changes and LOH across the genome for each sample, and if necessary, CNV data were manually centered with presumed diploid regions at copy level 2.0. Presumed diploid regions were considered to be those at the lowest non-LOH-region copy level, and the resulting presumed haploid and triploid regions obligately having LOH.

To determine whether a segmental chromosome alteration had occurred on any autosomal chromosome arm, the last 30 Mb of each chromosomal region (starting from the p- or q-terminus) of each segmental copy gain or deletion was analyzed (for chromosomes with arms less than 30 Mb in length, the entire chromosome arm was analyzed; for acrocentric chromosomes shorter segments of 15–25 Mb were queried; for 11q13.3 gains, the region of chromosome 11 from 68.5 Mb to 69.5 Mb in GRCh37 coordinates was used as this approximated the minimally gained region per Supplementary Fig. 8b). The copy level (non-logged) at 250 kb intervals was sampled within this 30 Mb range. A median level of 2.3 or higher (where 2.0 is diploid) among sampled regions was required to consider the arm gained; for copy losses, a value 0.3 copies or lower than the sample's median copy value was required. In addition, the median copy value for the entire opposing (control) arm had to be at least 0.3 less than that of the arm of interest for copy gains; for copy losses, the median copy value for the entire opposing arm had to be at least 0.3 above that of the arm of interest; these criteria enabled identification of segmental chromosome alterations and excluded whole-chromosome copy gains or losses. (However, for 11q13.3 gains, the control region was on the same

chromosome arm but further downstream—the last 30 Mb of chromosome 11—since 11q13.3 copy gains frequently extend to the p arm, making the p arm a poor control.)

For calling samples as having 9 or more whole-chromosome gains or not, the copy at each position was also sampled every 250 kb on each chromosome arm. If at least 80% of the chromosome arm had copy level of 2.25 or above, the arm was considered copy gained. If both arms of the chromosome met this criterion, the whole chromosome was considered gained. Samples with at least nine whole chromosomes gained were considered 9+ WC gain based on density plot analysis (Supplementary Fig. 5). In all, 59 samples had both WGS (based on CGI) and WES (Supplementary Fig. 1a), and these samples were used to test for concordance between WGS-based vs. WES-based arm-level copy alterations. WES-based arm-level copy calls with CnvKit had 86.1% sensitivity and 99.5% specificity relative to WGS-based copy calls (or 94.2% sensitivity and 98.3% specificity when analyzing only the 10 segmental driver gains and deletions shown in Fig. 1), indicating good accuracy.

**Structural variation analysis**. For Illumina WGS data, we used CREST (version 1.0)[70] to identify SVs after BWA alignment (version 0.5.9), and data curation was done manually afterwards. For CGI WGS data, we used SV data from our previous study, in which germline SVs were filtered out[11]. CNVs in each tumor were integrated into the structural variant analysis by matching breakpoints within a 5-kb window to rescue rearrangements with CNV support by manual curation. Samples were defined as positive for t(11;17) if they had a translocation joining 17q (at position 30 Mb or higher, GRCh37) to 11q (at between position 60 Mb to 80 Mb), as these were the hotspot translocation regions (Supplementary Fig. 8a).

**SNV and indel mutation identification**. Somatic SNVs and indels were called using Bambino (version 1.6)[71] for PCGP WGS, COG WES, and TARGET WES samples using GRCh37 as a reference genome after alignment with BWA (version 0.5.9), followed by postprocessing steps removing paralogous variants and artefacts[72]. For TARGET WGS data (generated by CGI), we used SNV and indel data from our previous study, in which we downloaded prior variant calls and filtered to generate clean SNV and indel results[11]. Significantly mutated SNVs and indels were identified with GRIN and MutSigCV[21,22]. *ATRX* mutation status was determined by either WGS or custom capture sequencing of the entire gene, including both exons and introns (COG cohort[57]). *FGFR1* N546K-mutant samples had *FGFR1* N546K VAFs of above 0.4 in samples with 2 or 4 copies of *FGFR1*, indicating clonal mutations. The only other patient with this variant (PATNKP) had *FGFR1* N546K VAFs of 0.32 and 0.22 in diagnosis and relapse samples, respectively, and 3 copies of *FGFR1*, indicating a likely clonal mutation.

**Artefactual variants**. We observed apparently somatic *DICER1* variants (primarily chromosome 14, position 95,596,418 in hg19 coordinates, G>T variants) in a small subset of samples sequenced by WES, but these variants could not be validated by capture validation and were considered artefacts. We also observed what appeared to be kataegis in the region of *MYCN* in 12 WGS samples. However, upon further inspection we found that most of these apparently somatic kataegis variants were in fact population variants, as 94% of them had a population allele frequency of 1% or above, compared to 0% of somatic variants in a control patient's non-kataegis variants ($P < 2.2 \times 10^{-16}$ by Fisher's exact test). They were likely contamination from another neuroblastoma sample with *MYCN* amplification run on the same sequencing lane, as the large amount of *MYCN* DNA in amplified samples may increase the potential for artefactual spill-over errors between samples run on the same lane.

**MYCN amplification status**. *MYCN* amplification status was based on three independent methods: (1) review of the clinical database where amplification was determined by southern blotting or by FISH, (2) WGS, and (3) Custom capture sequencing (COG cohort[57]). *MYCN* was considered copy-gained by WGS only when focal gains with $\log_2$ fold change of >2.0 relative to germline (~8 copies) were detected.

**RNA-seq data analysis**. Neuroblastoma RNA-Seq data were used from our previous study and were from TARGET[11]. Novel junctions were analyzed by RNApeg version 1[73]. For neuroblastoma differential gene expression analysis, 88 diagnosis samples with both WGS and RNA-Seq were divided into signature 18-positive ($n = 60$) and -negative ($n = 28$) groups based on the presence of any detectable signature 18. Genes with at least 1 count per million (CPM) in at least two samples were included, and RNA-Seq count data were transformed using voom[74] in R (which log-transforms data and performs statistical analysis preparatory to linear modeling), followed by the Limma[75] functions lmFit (for linear modeling), eBayes (for Bayesian differential expression analysis), and topTable (to report differentially expressed genes); default parameters were used for each of these functions, including Benjamini–Hochberg correction to obtain adjusted *P* values. A variant of this analysis was also performed with the inclusion of somatic *MYCN* alteration status (amplifications or point mutations were both considered positive) as a covariate to subtract potential effects of *MYCN* in the Limma differential expression analysis (Supplementary Fig. 19b) or with both *MYCN* and 17q gain status as

covariates (Supplementary Fig. 19c, d). ssGSEA[55] analysis was performed in R using 26 genes with neural function (based on the literature) which were statistically increased in expression in signature 18-negative samples (Supplementary Table 2) as an input gene set.

Rhabdomyosarcoma data are available through the St. Jude Cloud resource, including 31 PCGP and St. Jude Clinical Genomic samples with both WGS and RNA-Seq. Rhabdomyosarcoma RNA-Seq counts were obtained using HT-Seq. Rhabdomyosarcoma differential gene expression was performed using similar steps to the neuroblastoma analysis, except that (1) two batches were included as covariates since some samples utilized poly-A-based unstranded library preparation and some total RNA stranded library preparation, (2) the robust option was used with lmFit, due to the smaller sample size, to be robust against outliers, and (3) genes were more stringently filtered due to the smaller sample size, including genes with at least 10 CPM in at least 10 samples (including over 11,000 genes).

t-SNE clustering of neuroblastoma RNA-Seq samples was performed on 158 samples' TPM expression data using Seurat[76] in R.

**Mutual exclusivity and co-occurrence of mutations**. We tested mutual exclusivity and co-occurrence of recurrent mutations, using 182 diagnosis WGS samples, including only recurrent mutations mutated in five or more samples in that sample set. For each mutation pair A and B, we performed a two-sided Fisher's exact test according to their mutation status. We performed this analysis after removing non-independent interactions. SNVs and indels were always classified as independent from all other events, while translocations and copy number alterations were considered non-independent if (a) a structural variant joined them directly, as in the case of t(11;17) variants that directly joined to the end of an 11q13.3 gain and the beginning of a 17q gain (all three variants would be considered non-independent), or (b) structural variants joined the two variants through intervening chromosome sequences with sequential structural variants 15 Mb or less apart. Whenever such non-independent variants co-occurred in the same patient, the interaction was not included at all in the 2 × 2 Fisher's exact test table (Fig. 3a).

**Evolutionary timing of copy number occurrences**. To determine whether copy number gains were an early or late event in neuroblastoma evolution, we used mutations in regions with 3 copies (as this was the most frequent level of copy gain observed, and also is the easiest to model as it has fewer possible allelic configuration than higher copy gains) in 103 out of the 182 diagnosis samples. These 103 were selected due to their high tumor purity (at least 70%, calculated based on copy number and allele frequency data) and the presence of at least 20 mutations in 3-copy regions on any autosomal chromosome for analysis. The expected VAF for mutations on 2 of 3 copies (0.67 when tumor purity is 100%, or proportionally less if purity is lower) or 1 of 3 copies (0.33 when tumor purity is 100%) was calculated for each sample based on its purity, and the mean of the two values was considered a VAF cutoff (0.5 when tumor purity is 100%) between mutations on one 1 of 3 alleles (≤0.5 VAF), and mutations on 2 or more out of 3 alleles (>0.5 VAF). The percent of mutations in each sample in each of these two categories, for each sample's mutations which are in 3-copy regions, is shown in Supplementary Fig. 7. We assumed that somatic mutations accumulating before a copy gain occurred at an equal rate on the duplicated and non-duplicated chromosome, such that immediately after the duplication, half (50%) of previously generated somatic mutations were on 1 of 3 alleles, and the other half (50%) were on 2 of 3 alleles. Mutagenesis after the copy gain will occur on 1 of 3 copies, such that the percentage of mutations on 1 of 3 alleles will increase over time. When the molecular time prior to the copy gain (from fertilization to copy gain) and the molecular time after copy gain (from copy gain to tumor acquisition/sequencing) are equal, there were 75% of mutations on 1 of 3 alleles (two-thirds after the copy gain (50% overall) and one-third before (25% overall)) and 25% on 2 of 3 alleles (all before the copy gain). Most patients (85%) had more than 75% of mutations on 1 of 3 alleles, indicating that copy gains happened earlier than the majority of point mutations in these patients.

**Age associations with genetic alterations**. To compare the ages of diagnosis of patients with somatic alterations (including SNVs/indels, CNVs, or SVs) in *MYCN*, *TERT*, *ATRX*, or patients with none of the three variants, we generated the empirical cumulative distribution function using the ecdf function in R, using only diagnosis samples (Fig. 3d). To compare the frequency of all common genetic alterations between the three age groups, we compared the percent of patients in each group with each alteration using Fisher's exact test, using only diagnosis samples (Supplementary Fig. 15b and Fig. 3c). In Supplementary Fig. 15a, age of each sample with each alteration was analyzed as a continuous variable rather than by age groups, by Wilcoxon rank-sum test as described in the figure legend.

**Analysis of allele-specific expression**. We applied Cis-X[77] (version 1.4.0; software to discover novel structural or other somatic alterations associated with nearby allele-specific gene expression and thus potential cis-regulation) to determine whether *TERT* or *ALK* had germline allele-specific gene expression in sample PATDXC in the region near the t(2;5) translocation, using WGS and RNA-Seq

from this sample. This revealed a run of allele-specific gene expression (where only one allele of a heterozygous germline SNP is detected in RNA-Seq) near *TERT* despite both germline alleles being present by WGS (Supplementary Fig. 12b).

**Mutational signature analysis.** For the 205 neuroblastoma samples with WGS, we first determined the trinucleotide context of each somatic SNV, resulting in 96 possible mutation classes[41] to create a 205 × 96 (sample × mutation class) matrix containing the number of SNVs in each class in each sample. We then applied SigProfiler[44] (version 2.3.1) in MATLAB to extract mutational signatures de novo from the dataset. The optimal number of extracted signatures was 6, which had a signature stability above 0.95 and the lowest frobenius error of 1–6 signatures, while seven or more signatures had a signature stability of <0.8. We next tested which of these signatures was explained by the COSMIC v3 SNV signatures, signature definitions of which are described in the literature[44]. Five of the six extracted signatures were well-explained (cosine similarity ≥0.9) by some combination of the COSMIC v3 SNV signatures[44]; specifically substitution signatures (SBS in the referenced article) 1, 3, 5, 12, 18, 31, 40, 43, and 46 (hereafter queried signatures, since these were further analyzed). The sixth, T-10, was highly similar (cosine similarity of 0.947) to a likely artefactual signature, also referred to as T-10 in our previous study[11], associated with CGI library preparation; T-10 was included in our queried signatures to absorb artefactual signal and optimize signature detection.

We then used SigProfilerSingleSample (version 1.3) to test the presence of the queried signatures described above in 205 neuroblastoma samples with WGS, using the analysis_individual_samples function and the "signatures to be included in all samples regardless of rules or sparsity" parameter set to include substitution signatures 1, 3, and 5 as these are ubiquitous signatures (1 and 5)[46] or abundant specifically in neuroblastoma (3)[11] but potentially difficult to detect due to flat features. Signatures 12, 43, and 46, which were initially detected by de novo signature extraction, were not detected by SigProfilerSingleSample in any sample, leaving 7 signatures (including T-10) detected in our final analysis.

In all, 161 of 205 neuroblastoma samples' SNV catalogs were explained with cosine similarity ≥0.9 (comparing the original sample to the sample reconstructed by signatures) by the queried signatures (Supplementary Fig. 16). However, the samples with cosine similarity <0.9 had reliable signature data overall, as the cisplatin signature was only detected in relapsed cases regardless of cosine similarity (Supplementary Fig. 16), and the signature 18 proportion strongly correlated with the proportion of C>A mutations even when including samples with low cosine similarity ($r = 0.96$, $P = 1.38 \times 10^{-114}$ by Pearson correlation; note that other signatures detected in neuroblastoma do not substantially cause C>A mutations indicating that a strong signature 18-C>A correlation is expected)[44]. Thus the samples with cosine <0.9 were included in all analyses except the prediction of whether individual driver mutations were caused by a certain signature, which requires higher stringency. Most of the low-cosine samples were so due to a low mutation burden (≤200 mutations). Of the samples with cosine similarity <0.9, 32 of 46 (69.6%) had ≤200 mutations; in contrast, among included samples, only 3 of 159 (1.9%) had ≤200 mutations.

To identify variants likely caused by specific mutational signatures, we used an approach[62] which we have implemented previously[63], as follows. The probability that an SNV was caused by a specific signature was calculated as follows. Let $s_k$ represent the signature strength vector for a given sample (measured in number of SNVs caused by the signature), where $k = 1, 2, \ldots, 7$ is one of the 7 signatures detected (Supplementary Fig. 16), such that $\sum s_k$ equals the total number of SNVs in the sample. Let $c = 1, 2, \ldots, 96$ represent each of the 96 possible trinucleotide mutation types. Each signature mutates these 96 trinucleotide contexts with a probability $P_{c,k}$, where $\sum_c P_{c,k} = 1$. The probability that a mutation of interest $m$ (at trinucleotide context $c$) was caused by a specific signature $i$ is calculated as:

$$P(i|m) = \frac{s_i^* P_{c,i}}{\sum_{k=1}^{7} \left( s_k^* P_{c,k} \right)} \tag{1}$$

The numerator represents the number of mutations caused by a specific signature $i$ at the mutation context of interest, while the denominator represents the total number of mutations caused by all signatures detected in the sample at the mutation context of interest. Example calculations for a specific mutation are shown in Supplementary Fig. 21.

The rhabdomyosarcoma mutational signature analysis was performed by first testing for the presence of all of the COSMIC v3 signatures among 831 adult cancers sequenced by WGS through TCGA spanning 23 cancer types, using SNV data downloaded through the ICGC data portal; and among 1603 pediatric cancers with WGS spanning 39 cancer types, which have been analyzed as part of the PCGP[60] or St. Jude Clinical Genomics programs (available on St. Jude Cloud). The pediatric cancer data use the GRCh38 reference genome. Signature scores were obtained using SigProfilerSingleSample. After finding that rhabdomyosarcoma was the only solid tumor type where at least 20% of samples were signature 18-positive and where sufficient signature 18-positive samples were available, we focused further analysis on mutational signatures in this cancer type. Chromosome 8 gain status was obtained by performing copy analysis on these rhabdomyosarcomas using CONSERTING version 1.0[68].

**Reporting summary**. Further information on research design is available in the Nature Research Reporting Summary linked to this article.

## Data availability

WGS data in the PCGP cohort can be obtained from EGA using accession EGAS00001000213. WGS, WES, and RNA-seq data from TARGET can be accessed from dbGaP via accession phs000218. COG WES bam files which are new to this study ($n = 317$ samples) are available on EGA, under controlled access based on approval of the PCGP Steering Committee (PCGP_data_request@stjude.org) in accordance with the community practice for human genomic data protection, at accession EGAD00001005484, where bam file names ending _D1.bam indicate tumor (diagnosis) samples and bam file names ending _G1.bam indicate germline samples. COG USI patient identifiers associated with each bam file, and the specific repository containing each patient's raw data, can be found in Supplementary Data 1. All somatic alterations identified are recorded in Supplementary Data 1–6 and can also be viewed interactively using ProteinPaint[78] at https://pecan.stjude.cloud/proteinpaint/study/PanNeuroblastoma.Alterations. Source data are provided with this paper. Any remaining data are available within the Article, Supplementary files, or are available from the authors upon request.

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

## Acknowledgements

We thank the Children's Oncology Group for providing clinical information for samples analyzed in this study. This research was supported by the National Cancer Institute (NCI) of the National Institutes of Health through Cancer Center Support Grant P30 CA021765, by NCI R01 CA216391-01A1 (J.Z.), by NCI R35 CA220500 (J.M.M.), by NCI U10 CA180899 to the Children's Oncology Group Statistics and Data Center (A.N.), and by the American Lebanese Syrian Associated Charities of St. Jude Children's Research Hospital. The content is solely the responsibility of the authors and does not necessarily represent the official views of the National Institutes of Health. We thank Dr. Elli Papaemmanuil for valuable information regarding artefactual variants.

## Author contributions

J.R.D. and J.Z. designed the study. Data analysis (including statistical and computational analysis) was performed by S.W.B., Y.L., X.M., K.H., A.M.G., M.M., X.C., M.R., S.L., E.M.D., C.C., X.Z., J.W., A.N., and J.Z. Material support was provided by M.D.H., M.A.D., N.V.C., and J.M.M. Genomic sequencing of patient tissue was performed or overseen by J.E., H.L.M., J.N., and L.W. The manuscript was written by S.W.B. and J.Z. with critical feedback from M.A.D., M.D.H., and J.M.M. The study was supervised by M.D.H., M.A.D., and J.Z.

## Competing interests

N.V.C. reports receiving commercial research grants from Y-mabs Therapeutics and Abpro-Labs Inc., holding ownership interest/equity in Y-Mabs Therapeutics Inc., holding ownership interest/equity in Abpro-Labs, and owning stock options in Eureka Therapeutics. N.V.C. is the inventor of issued and pending patents filed by Memorial Sloan-Kettering Cancer Center (MSK), including those licensed by MSK to Ymabs Therapeutics, Biotec Pharmacon, and Abpro-labs. Both MSK and N.V.C. have financial interest in Y-mabs. N.V.C. is an advisory board member for Abpro-Labs and Eureka Therapeutics. The remaining authors declare no competing interests.
