## [Peer Review File · Nature Communications]

REVIEWER COMMENTS

Reviewer #1 (Remarks to the Author):

The authors have adequately handled most of the reviewers comments, however a major concern that still holds is the limited amount of novel data presented in the manuscript. Most of the presented data/results (although now tested on a larger samples cohort) have been published before.

Concerning comment 5 of reviewer 1: the added sentence in the manuscript should be rephrased as it is very unclear.

According to Figure 1, several samples present without any genomic aberration (copy number or mutation): what is the percentage of tumor cells in these cases?

Reviewer #2 (Remarks to the Author):

Thank you for asking me to re-review the paper, this time in a different Nature journal. The authors now seem to acknowledge that most of their observations merely confirm known features of the neuroblastoma genome. There remain specific technical and semantic concerns that I have (see discussion further below). Leaving those aside, the issue of novelty, or rather the lack thereof, remains. In the introduction of their rebuttal, the authors highlight three findings that they consider to be novel:

A) "Our study provides a comprehensive view of the driver alteration landscape in age-dependent subgroups which had only been reported at a smaller scale in both the sample number and number of events." – What the authors have achieved is to increase the confidence interval around the frequency of drivers. I would not consider this to be a substantial shift in our understanding of the drivers of neuroblastoma.

B) "FGFR1 is a known cancer gene but its role as a neuroblastoma driver gene is first established through this study as previous studies only reported a singleton event, thus lacking the statistical power to assess the significance of this alteration." – The authors still do not acknowledge in their paper that this mutation has been described before. Furthermore, there is no mentioning of the fact amplification of FGFR3 is a known feature of olfactory neuroblastoma. Implicating FGFR signalling in neuroblastoma would therefore not appear to be a novel revelation. Even if this observation were novel, it would still only be relevant in a tiny proportion of tumours (1%). It does not constitute a significant shift in our understanding of the driver mutations underpinning neuroblastoma.

C) "We show for the first time the association of mitochondrial gene expression to mutational signature 18 in neuroblastoma, a significant finding on the potential mutagenic processes of neuroblastoma." – There remain several issues with this association; mainly that it is not adjusted for MYCN amplification. As MYCN amplified tumours have more signature 18 mutations, they are likely to proliferate faster and thus have more mitochondrial usage. Therefore, I suspect that the association will disappear once adjusted for MYCN status. Furthermore, this finding is somewhat

undermined by the recent publication of this paper here from the authors - (<https://www.nature.com/articles/s41467-020-14682-6>) - which circumscribes the association of mitochondrial genes expression and ROS-associated mutations without using the word signature 18.

Additional points (numbering as per rebuttal):

1.3. I think the authors' language ought to be much stronger. These changes have not just been previously identified as "recurrent drivers". These are fundamental features of the neuroblastoma genome that determine risk and thus treatment.

3.1. I appreciate the authors' efforts here. How did the authors determine whether rearrangements were independent or not? Was this done informatically or manually? The issue that remains is that none of the associations, as the authors highlight themselves, is novel. In particular the mutual exclusivity of MYCN and ATRX has now been reported on by the authors themselves (<https://www.nature.com/articles/s41467-020-14682-6>).

4.1. This is an odd response. FOXO1 fusions are canonical, disease defining events in alveolar RMS, which are typically caused by balanced translocations (mostly) without a wildly rearranged genome. I am not sure what this has to do with the point I raised.

Based on the numbers the authors present it would seem important that they highlight in the manuscript that their classification of NB by age is, although broadly speaking "ok", imprecise. A stage 4 tumour should not be lumped together with low risk tumours in Group A. This needs to be highlighted to the reader. The numbers also highlight what the problem was. This study is not large enough to demonstrate that SHANK2 rearrangement are significantly less prevalent in low risk tumours which are generally less rearranged than high risk tumours. 1

4.2. I disagree with this line of reasoning, mostly because I think some assumptions are invalid. Perhaps let us simplify the discussion here. Can the authors assess the enrichment of PTPRD breakpoints across all samples constructing a statistical model that adjusts for the following confounders: age; MYCN status; rearranged-ness (diploid/tetraploid VS everything else). I suspect that they will not find an enrichment of PTPRD variants in any group.

5.2. It would seem that the authors have not fully appreciated the study that I cited. That particular experiment went through rounds of subcloning individual cells to segregate culture artefacts from signature 18 mutations generated by culture-independent mutational processes.

5.4. The issue remains that what the authors say that have found, as association of MYCN alterations with signature 18, has been shown before, as the authors themselves recognise (see reference 80 they quoted): "MYCN-altered samples had significantly more signature 18 (Fig. 4E), consistent with recently reported MYCN-induced ROS generation in neuroblastoma."

6.1. As I was writing the paragraph initially, I was wondering whether the authors would rebut this using the well known anomaly of BRAF V600E mutations. What I am asking the authors to do is to statistically assess whether signature 18 is causing more driver events that one would expect to find by chance, given the prevalence of signature 18 mutations in each sample. The authors need to

consider in their model the actual trinucleotide context and amino acid change required to generate the mutation in question. It would be helpful if the authors were able to rebut my point with data analyses.

Point-by-point response

We thank reviewer #1 and #2 for taking the time to review our revised manuscript “Pan-neuroblastoma analysis reveals age- and signature-associated driver alterations” (NCOMMS-20-08286A). We have addressed each comment as detailed below and revised the manuscript accordingly using the Track Changes feature.

Reviewer #1 (Remarks to the Author):

The authors have adequately handled most of the reviewers comments, however a major concern that still holds is the limited amount of novel data presented in the manuscript. Most of the presented data/results (although now tested on a larger samples cohort) have been published before.

Concerning comment 5 of reviewer 1: the added sentence in the manuscript should be rephrased as it is very unclear.

[Response]: Our manuscript is a pan-cancer analysis of three major subgroups of neuroblastoma which compares the mutational patterns in the three subgroups defined by age of onset. By integrating data generated from previous studies with an additional >300 exome cases generated from this study, we were able to determine the prevalence of genetic alterations across the three age groups, which will serve as an important resource for the broad research community.

Regarding the specific comment 5, we apologize for the lack of clarity and agree that the statement should be re-worded. The unclear statement was as follows:

“Further, *FGFR1* expression was above-median for all genes across the cohort among the 169 samples analyzed by RNA-Seq (61st percentile, or a median transcripts per million (TPM) of 16.3), including in the two *FGFR1*-mutant samples that also had RNA-Seq, indicating that the gene is expressed in neuroblastoma.”

We have updated this to the following to state the *FGFR1* expression in the *FGFR1*-mutant samples specifically:

“Further, the median *FGFR1* expression was ranked at the 61st percentile of expression in this cohort (median transcripts per million (TPM) of 16.3), indicating that *FGFR1* is expressed in neuroblastoma. Specifically, in the two *FGFR1*-mutant samples that also had RNA-Seq, *FGFR1* expression was at 79.6 TPM (the 87th percentile of expression within the sample) or 21.6 TPM (69th percentile), and the mutant *FGFR1* alleles were expressed at similar allele frequencies as in DNA (0.30 VAF in RNA vs. 0.22 in WGS, in one example patient).”

Location(s) of changes: Results section “Kinase alterations in *FGFR1* and truncated *ALK* variants” (lines 199-204).

According to Figure 1, several samples present without any genomic aberration (copy number or mutation): what is the percentage of tumor cells in these cases?

[Response]: A total of 136 samples (out 685) lacked recurrent SNV, indel, structural or copy alterations reported in Fig. 1, and also had no whole-chromosome copy alterations. We were able to collect pathology review for 130 of these cases and found that tumor purity was 60% or above for all cases. These cases were enriched in low-stage tumors. We added the following text in the revision

“Of the 685 samples with WGS or WES, 136 (20%) lacked any of these recurrent alterations and had no whole-chromosome copy alterations. These 136 samples were enriched in low disease stage (only 19% were stage 4 compared to 70% of other samples, $P < 2.2 \times 10^{-16}$) and their paucity of somatic alterations was not caused by low tumor purity—pathology review, available for 130 out of the 136 samples, showed that tumor purity exceeded 60% in all cases.”

Location(s) of changes: Results section “Landscape of somatic mutations in neuroblastoma” (lines 160-165).

Reviewer #2 (Remarks to the Author):

Thank you for asking me to re-review the paper, this time in a different Nature journal. The authors now seem to acknowledge that most of their observations merely confirm known features of the neuroblastoma genome. There remain specific technical and semantic concerns that I have (see discussion further below). Leaving those aside, the issue of novelty, or rather the lack thereof, remains. In the introduction of their rebuttal, the authors highlight three findings that they consider to be novel:

A) “Our study provides a comprehensive view of the driver alteration landscape in age-dependent subgroups which had only been reported at a smaller scale in both the sample number and number of events.” – What the authors have achieved is to increase the confidence interval around the frequency of drivers. I would not consider this to be a substantial shift in our understanding of the drivers of neuroblastoma.

[Response]: Our manuscript is a pan-neuroblastoma analysis of three major subgroups of this disease, which compares the mutational patterns in the three subgroups defined by age of onset. The sample size is an important factor in ensuring the rigor of analysis so that the harmonized data and the results can be an important resource for the research community.

B) “FGFR1 is a known cancer gene but its role as a neuroblastoma driver gene is first established through this study as previous studies only reported a singleton event, thus lacking the statistical power to assess the significance of this alteration.” – The authors still do not acknowledge in their paper that this mutation has been described before. Furthermore, there is no mentioning of the fact amplification of FGFR3 is a known feature of olfactory neuroblastoma. Implicating FGFR signalling in neuroblastoma would therefore not appear to be a novel revelation. Even if this observation were novel, it would still only be relevant in a tiny proportion of tumours (1%). It does not constitute a significant shift in our understanding of the driver mutations underpinning neuroblastoma.

[Response]: The reviewer states that “The authors still do not acknowledge in their paper that this mutation [*FGFR1*] has been described before.” However, both our original and revised manuscripts cite several studies shown that *FGFR1* mutations have been reported as recurrent in other cancer types, and in a single neuroblastoma tumor, including the following statement:

“*FGFR1* N546K was previously reported in a single neuroblastoma patient¹ and therefore has not been considered a driver gene in neuroblastoma. This variant activates MAPK signaling in functional studies from other tumor types^{2,3}, and is recurrent in pediatric low-grade glioma^{4,5}, indicating it is a driver mutation.”

Olfactory neuroblastoma (or esthesioneuroblastoma), despite its misleading name, is a clinically and molecularly distinct entity from neuroblastoma. Olfactory neuroblastomas arise from sensory olfactory neuroepithelium⁶⁻⁸ and occur most frequently in adults⁹, whereas neuroblastoma is of neuroendocrine origin¹⁰ and occurs primarily in children¹¹. The histology, epidemiology, staging and therapy likewise differ. Further, olfactory neuroblastomas are molecularly distinct from neuroblastoma, lacking *MYCN*¹² or

ALK alterations¹³. Therefore, we feel that *FGFR3* amplification in this distinct tumor entity is not relevant to neuroblastoma, and discussing it in the context of our findings would lead to confusion in the narrative.

C) “We show for the first time the association of mitochondrial gene expression to mutational signature 18 in neuroblastoma, a significant finding on the potential mutagenic processes of neuroblastoma.” – There remain several issues with this association; mainly that it is not adjusted for *MYCN* amplification. As *MYCN* amplified tumours have more signature 18 mutations, they are likely to proliferate faster and thus have more mitochondrial usage. Therefore, I suspect that the association will disappear once adjusted for *MYCN* status. Furthermore, this finding is somewhat undermined by the recent publication of this paper here from the authors - (<https://www.nature.com/articles/s41467-020-14682-6>) - which circumscribes the association of mitochondrial genes expression and ROS-associated mutations without using the word signature 18.

[Response]: We appreciate this suggestion. In our revised manuscript we have performed the differential gene expression analysis while including *MYCN* alteration status as a covariate, thus adjusting for its effects (new Supplementary Fig. 18B). The mitochondrial gene expression remained statistically significant after this adjustment, and we have also shown the expression of a few example mitochondrial genes in signature 18-positive vs. -negative samples *including only samples lacking MYCN alterations* to show that the mitochondrial expression increase remains regardless of *MYCN*. This is consistent with the fact that several of the mitochondrial genes are found on chromosome 17q, and 17q gains are associated with signature 18 independent of *MYCN* (Supplementary Fig. 18A, which was included in both the previous and revised manuscripts).

Further, when including signature 18 status, *MYCN* status, and 17q gain status all as covariates, signature 18 was not significantly associated with the expression of any gene, and 17q gains but not *MYCN* were associated with increased mitochondrial gene expression (new Supplementary Fig. 18C, D). This indicates that the link between signature 18 and mitochondrial gene expression is due to 17q gains and not *MYCN*. We have included this analysis in the revised manuscript. We note here that *MYCN*'s primary effect on ROS is through inducing glutaminolysis, which depletes ROS-protective glutathione, as we and others have shown^{14,15}. Thus, 17q gains may increase mitochondrial ROS through gains of mitochondrial genes, while *MYCN* decreases protection from ROS, which could plausibly explain the additive effect of the two alterations on signature 18 (Supplementary Fig. 18A).

Regarding the study noted by the reviewer, we note that it was cited in our previous version of the manuscript as follows:

“*MYCN*-altered samples had significantly more signature 18 (Fig. 4E), consistent with recently reported *MYCN*-induced ROS generation in neuroblastoma¹⁵.”

We would like to point out that the *ATRX-MYCN* exclusivity study, published by our group, is a functional study which did not perform any mutational signature analysis. Specifically, it did not make any reference to ROS-associated mutagenesis nor signature 18, but focused only on the DNA damage response associated with *MYCN*-induced ROS. Further, as the reviewer noted during the previous revision, signature 18 and ROS have not been definitively linked in neuroblastoma; thus, our finding that *MYCN* is associated with signature 18 is distinct from (though potentially related to) that study's focus on *MYCN*-induced ROS. Finally, the association of 17q gain with signature 18, including the expression of mitochondrial genes found on 17q (*ATP5G1*, *ICT1*, and *MRPS7*, for example), has not been noted in the study in question nor any previous study of which we are aware.

Locations of change(s): Results section “Mutational signature 18 is associated with increased mitochondrial gene expression, 17q gains, and *MYCN*” (Lines 336-345), Supplementary Fig. 18.

Additional points (numbering as per rebuttal):

1.3. I think the authors' language ought to be much stronger. These changes have not just been previously identified as "recurrent drivers". These are fundamental features of the neuroblastoma genome that determine risk and thus treatment.

[Response]: We note that not all recurrent alterations shown in Fig. 1 are associated with risk and treatment. Of the lesions shown, only segmental copy number changes in chromosomes and *MYCN* are used to routinely determine risk group designation. Only *ALK* is used currently to select specific therapeutics, and while *TERT* alterations are shown to correlate with poor risk, they have not been incorporated into current risk classification schemas. Other alterations identified, including *SHANK2* and *PTPRD*, are not used for clinical risk classification, nor are they associated with survival, as we have shown. Therefore, we do not feel it is warranted to make the statement suggested by the reviewer in this section, which is focused on understanding the recurrent mutational landscape of neuroblastoma.

The information regarding the prognosis of specific driver alterations has already been noted at other more appropriate locations in the manuscript, including the Introduction, and our systematic analysis of survival correlations in the Results section. Please note, for example, the following statements:

1. From the Introduction: "Whole chromosome gains are frequently observed in low-risk neuroblastoma¹⁶, while gains or losses of chromosome arms (segmental chromosome alterations), including loss of 1p, 3p, 4p and 11q and gain of 1q, 2p and 17q^{17,18}, are associated with poor prognosis¹⁹. *MYCN* amplification is the most frequent driver in neuroblastoma, occurring in ~20% of cases and conferring poor prognosis²⁰⁻²²."
2. From the Results: "*MYCN*, *ALK*, *ATRX*, and Ras pathway alterations; segmental deletion of 1p, 3p, and 11q; and segmental gain of 1q, 2p, 7q, 11q13.3, 12q, and 17q were each significantly associated with poor overall survival (Supplementary Fig. 9), consistent with previous reports^{20,23-27}."
3. From the Results: "*FGFR1* and *TERT* alterations trended towards an association with worse survival but were not statistically significant ($P = 0.09$ and $P = 0.1$, respectively; Supplementary Fig. 9). In previous studies, *TERT* has been associated with both worse survival²⁸ and no difference in survival²⁹. These reported findings, together with ours, suggest that *TERT* alterations alone have a modest effect on patient outcomes."

The last-noted statement about *TERT* alterations shows that nuance is needed in describing the prognostic significance of each alteration. Thus, we feel that making a statement that all Fig. 1 driver alterations are associated with prognosis and treatment would be inaccurate, repetitive of information presented in the Introduction, untimely in a section focused only on the genomic landscape, and deserves its own detailed Results description as we have done.

3.1. I appreciate the authors' efforts here. How did the authors determine whether rearrangements were independent or not? Was this done informatically or manually? The issue that remains is that none of the associations, as the authors highlight themselves, is novel. In particular the mutual exclusivity of *MYCN* and *ATRX* has now been reported on by the authors themselves (<https://www.nature.com/articles/s41467-020-14682-6>).

[Response]: This analysis was done manually by verifying the computational predictions generated from WGS. Using visualization tools that we developed^{30,31}, we evaluated the junction reads across the structural variation breakpoints as well as read-depth changes in computationally predicted SVs and CNVs for every patient. This enabled us identify whether CNVs and SVs were joined: (a) directly (where

a translocation directly joins the termini of two copy alterations, for example) or (b) through intervening chromosome sequences with sequential structural variants 15 Mb or less apart. Details are provided in the Methods section.

As an example of (b), in one patient a 1p deletion and a 17q gain were apparently linked together, but not directly—an intervening segment of chromosome 1q was a likely bridge between the two. Since all three breakpoints (1p to 1q to 17q) could be joined in a theoretical consensus contig with the intervening 1q sequence less than 15 Mb in length, we considered that the 1p deletion and 17q gain were likely linked and removed the co-mutation of the two in that patient from the statistical calculation.

4.1. This is an odd response. FOXO1 fusions are canonical, disease defining events in alveolar RMS, which are typically caused by balanced translocations (mostly) without a wildly rearranged genome. I am not sure what this has to do with the point I raised.

Based on the numbers the authors present it would seem important that they highlight in the manuscript that their classification of NB by age is, although broadly speaking “ok”, imprecise. A stage 4 tumour should not be lumped together with low risk tumours in Group A. This needs to be highlighted to the reader. The numbers also highlight what the problem was. This study is not large enough to demonstrate that SHANK2 rearrangements are significantly less prevalent in low risk tumours which are generally less rearranged than high risk tumours. 1

[Response]: To address the reviewer’s concern, we have analyzed driver variant/age group associations among stage 4 samples only, in addition to all samples (revised Supplementary Fig. 15B). When doing this, the *PTPRD*, t(11;17), 2p gain, and Ras pathway age associations became non-significant, while all other age associations noted in the manuscript (*MYCN*, *ATRX*, *TERT*, 1p deletion, etc.) remained statistically significant. We have noted this in the revised manuscript.

Locations of changes(s): Results section “Age-related genomic aberrations” (lines 258-271).

4.2. I disagree with this line of reasoning, mostly because I think some assumptions are invalid. Perhaps let us simplify the discussion here. Can the authors assess the enrichment of *PTPRD* breakpoints across all samples constructing a statistical model that adjusts for the following confounders: age; *MYCN* status; rearranged-ness (diploid/tetraploid VS everything else). I suspect that they will not find an enrichment of *PTPRD* variants in any group.

[Response]: As noted in response to the previous comment, we have analyzed driver variant/age group associations among stage 4 samples only, in addition to all samples (revised Supplementary Fig. 15B). As hypothesized by the reviewer, the *PTPRD* age associations became non-significant as there were only 11 group A (<1.5 years) samples with WGS. We have noted this in the revised manuscript, and the revised statement reads as follows:

“By contrast, *PTPRD* genetic alterations (consisting of gene-disrupting SVs and focal deletions) were significantly higher in groups B and C as they were completely absent from group A (Supplementary Fig. 15A, B), and Ras pathway mutations were enriched in group B, *although there was no significant age group difference for PTPRD and Ras pathway mutations when including only stage 4 samples, suggesting their age specificity was related to higher disease stage* (Supplementary Fig. 15B).”

Locations of changes(s): Results section “Age-related genomic aberrations” (lines 258-260).

5.2. It would seem that the authors have not fully appreciated the study that I cited. That particular experiment went through rounds of subcloning individual cells to segregate culture artefacts from signature 18 mutations generated by culture-independent mutational processes.

[Response]: We respectfully disagree that subcloning cells in culture is able to “segregate culture artefacts from...culture-independent mutational processes” and we see no claim in the study³² to this

effect. Cell culture artefacts will continue to be generated in cell culture models regardless of the experimental manipulation performed. Indeed, the study in question³² shows that signature 18 was generated in cell lines from tissues not normally generating signature 18, even when performing subcloning. For example, their Figure 3 shows that BT474 and AU565 breast cancer cell lines generate signature 18, while patient breast tumors do not usually generate signature 18³³. Further, the authors state that while signature 18 was detected continuously in the two neuroblastoma cell lines studied using their sequential subcloning experiment (NB13 and BE2-M17), signature 18 was also observed as a feature of cell culture in general in the following direct quotation (underline added):

“SBS18 [Signature 18] is prominent in neuroblastoma (Alexandrov et al., 2013a) and continued to be generated in all of the neuroblastoma cell lines examined (Figure 3). It was also, however, observed in many daughter clones that were whole-genome sequenced (and thus captured sufficient numbers of mutations) of cell lines in which it was not detected in stocks (Figure 3). It therefore appears to be a common feature of *in vitro* culture, as previously noted (Rouhani et al., 2016). SBS18 may be generated by DNA damage caused by reactive oxygen species (Viel et al., 2017), and this mechanism could plausibly mediate its manifestation as a consequence of *in vitro* cell culture³².”

Further, another study³⁴ analyzing signatures in iPS cells treated with various perturbations also noted that signature 18 is generated under basal cell culture conditions, as follows:

“Last, the ubiquitous background signature present across the control [single-cell iPS clones sequenced after solvent control treatment] samples is similar to COSMIC Signature 18, previously hypothesized to be due to ROS³⁴.”

Therefore, we feel it worthwhile to show that signature 18 is generated continuously in actual patient tumors, which complements this study of only two neuroblastoma cell lines which explicitly states that signature 18 is a common feature of cell culture, including when performing single-cell subcloning. We maintain that the following statement in our manuscript is valid, and the use of the term “suggested” is warranted, given the caveats stated in the study³² itself, and its use of only two neuroblastoma cell lines:

“We found that signature 18 may be both an early event in neuroblastoma, causing truncal mutations, and an on-going mutational event causing relapse-specific mutations, based on analysis of five patients with matched diagnosis and relapse samples (Fig. 4B), as suggested previously by cultured neuroblastoma cell models³².”

5.4. The issue remains that what the authors say that have found, as association of MYCN alterations with signature 18, has been shown before, as the authors themselves recognise (see reference 80 they quoted): “MYCN-altered samples had significantly more signature 18 (Fig. 4E), consistent with recently reported MYCN-induced ROS generation in neuroblastoma.”

[Response]: As noted by the reviewer during the previous revision, signature 18 has not been explicitly linked to ROS in neuroblastoma. Nor did the study in question¹⁵ analyze *MYCN*-induced mutagenesis of signature 18, but only ROS generation and the resulting DNA damage response¹⁵. Further, the main novelty of the signature 18 correlations is to show that 17q gains (and increased expression of 17q mitochondrial genes), is associated with increased signature 18, independent of *MYCN* status (Supplementary Fig. 18).

6.1. As I was writing the paragraph initially, I was wondering whether the authors would rebut this using the well known anomaly of BRAF V600E mutations. What I am asking the authors to do is to statistically assess whether signature 18 is causing more driver events that one would expect to find by chance, given the prevalence of signature 18 mutations in each sample. The authors need to consider in their model the

actual trinucleotide context and amino acid change required to generate the mutation in question. It would be helpful if the authors were able to rebut my point with data analyses.

[Response]: In the revised manuscript, we have noted that there is no significant difference between the percentage of driver mutations caused by signature 18 (52%) compared to the percentage of all mutations genome-wide which were caused by signature 18 (56%). The updated statement is as follows:

“The percent of driver SNVs most likely induced by signature 18 (52%) was similar to the percentage of all mutations caused by signature 18 across the 38 samples analyzed (56%, $P = 0.64$ by Fisher’s exact test), indicating that signature 18 was proportionally likely to cause driver SNVs as SNVs in general. This indicates that signature 18 is likely a driver of disease progression in neuroblastoma, in contrast with passenger mutational signatures, such as the kataegis-associated APOBEC signature in osteosarcoma which causes no known driver SNVs in that cancer type³⁵.”

Location(s) of changes: Results section “Driver mutations associated with signature 18” (lines 392-397).

References

1. Eleveld, T. F. *et al.* Relapsed neuroblastomas show frequent RAS-MAPK pathway mutations. *Nat. Genet.* **47**, 864–871 (2015).
2. Rivera, B. *et al.* Germline and somatic FGFR1 abnormalities in dysembryoplastic neuroepithelial tumors. *Acta Neuropathol.* **131**, 847–863 (2016).
3. Bennett, J. T. *et al.* Mosaic Activating Mutations in FGFR1 Cause Encephalocraniocutaneous Lipomatosis. *Am. J. Hum. Genet.* **98**, 579–587 (2016).
4. Chiang, J. C. & Ellison, D. W. Molecular pathology of paediatric central nervous system tumours. *J. Pathol.* **241**, 159–172 (2017).
5. Jones, D. T. W. *et al.* Recurrent somatic alterations of FGFR1 and NTRK2 in pilocytic astrocytoma. *Nat. Genet.* **45**, 927–932 (2013).
6. Dulguerov, P., Allal, A. S. & Calcaterra, T. C. Esthesioneuroblastoma: A meta-analysis and review. *Lancet Oncology* **2**, 683–690 (2001).
7. Faragalla, H. & Weinreb, I. Olfactory neuroblastoma: A review and update. *Adv. Anat. Pathol.* **16**, (2009).
8. Shah, K. & Perez-Ordóñez, B. Neuroendocrine Neoplasms of the Sinonasal Tract: Neuroendocrine Carcinomas and Olfactory Neuroblastoma. *Head Neck Pathol.* **10**, 85–94 (2016).
9. Bell, D. *et al.* Prognostic Utility of Hyams Histological Grading and Kadish-Morita Staging Systems for Esthesioneuroblastoma Outcomes. *Head Neck Pathol.* **9**, 51–59 (2015).
10. Howman-Giles, R., Shaw, P. J., Uren, R. F. & Chung, D. K. V. Neuroblastoma and Other Neuroendocrine Tumors. *Seminars in Nuclear Medicine* **37**, 286–302 (2007).
11. Maris, J. M. Recent advances in neuroblastoma. *N. Engl. J. Med.* **362**, 2202–11 (2010).
12. Thompson, L. D. R. Olfactory Neuroblastoma. *Head Neck Pathol.* **3**, 252–259 (2009).
13. Lazo de la Vega, L. *et al.* Comprehensive molecular profiling of olfactory neuroblastoma identifies potentially targetable FGFR3 amplifications. *Mol. Cancer Res.* **15**, 1551–1557 (2017).
14. Wang, T. *et al.* MYCN drives glutaminolysis in neuroblastoma and confers sensitivity to an ROS augmenting agent article. *Cell Death Dis.* **9**, 1–12 (2018).
15. Zeineldin, M. *et al.* MYCN amplification and ATRX mutations are incompatible in neuroblastoma. *Nat. Commun.* **11**, 913 (2020).
16. Schleiermacher, G. *et al.* Chromosomal CGH identifies patients with a higher risk of relapse in neuroblastoma without MYCN amplification. *Br. J. Cancer* **97**, 238–246 (2007).
17. Pugh, T. J. *et al.* The genetic landscape of high-risk neuroblastoma. *Nat. Genet.* **45**, 279–284 (2013).
18. Coco, S. *et al.* Age-dependent accumulation of genomic aberrations and deregulation of cell cycle

- and telomerase genes in metastatic neuroblastoma. *Int. J. Cancer* **131**, 1591–1600 (2012).
19. Defferrari, R. *et al.* Influence of segmental chromosome abnormalities on survival in children over the age of 12 months with unresectable localised peripheral neuroblastic tumours without MYCN amplification. *Br. J. Cancer* **112**, 290–295 (2015).
 20. Maris, J. M., Hogarty, M. D., Bagatell, R. & Cohn, S. L. Neuroblastoma. *Lancet* **369**, 2106–2120 (2007).
 21. Irwin, M. S. & Park, J. R. Neuroblastoma: paradigm for precision medicine. *Pediatr. Clin. North Am.* **62**, 225–56 (2015).
 22. Matthay, K. K. *et al.* Neuroblastoma. *Nat. Rev. Dis. Prim.* **2**, 16078 (2016).
 23. De Brouwer, S. *et al.* Meta-analysis of Neuroblastomas Reveals a Skewed ALK Mutation Spectrum in Tumors with MYCN Amplification. *Clin. Cancer Res.* **16**, 4353–4362 (2010).
 24. Kurihara, S., Hiyama, E., Onitake, Y., Yamaoka, E. & Hiyama, K. Clinical features of ATRX or DAXX mutated neuroblastoma. *J. Pediatr. Surg.* **49**, 1835–1838 (2014).
 25. Bown, N. *et al.* Gain of Chromosome Arm 17q and Adverse Outcome in Patients with Neuroblastoma. *N. Engl. J. Med.* **340**, 1954–1961 (1999).
 26. Attiyeh, E. F. *et al.* Chromosome 1p and 11q Deletions and Outcome in Neuroblastoma. *N. Engl. J. Med.* **353**, 2243–2253 (2005).
 27. Spitz, R., Hero, B., Ernestus, K. & Berthold, F. Deletions in chromosome arms 3p and 11q are new prognostic markers in localized and 4s neuroblastoma. *Clin. Cancer Res.* **9**, 52–8 (2003).
 28. Valentijn, L. J. *et al.* TERT rearrangements are frequent in neuroblastoma and identify aggressive tumors. *Nat. Genet.* **47**, 1411–1414 (2015).
 29. Suo, C. *et al.* Accumulation of potential driver genes with genomic alterations predicts survival of high-risk neuroblastoma patients. *Biol. Direct* **13**, 14 (2018).
 30. Edmonson, M. N. *et al.* Bambino: a variant detector and alignment viewer for next-generation sequencing data in the SAM/BAM format. *Bioinformatics* **27**, 865–866 (2011).
 31. Liu, Y. *et al.* Abstract 1287: Exploring somatic DNA structural alteration and aberrant genomic interactions in cancer through GenomePaint. in *Cancer Research* **78**, 1287–1287 (American Association for Cancer Research (AACR), 2018).
 32. Petljak, M. *et al.* Characterizing Mutational Signatures in Human Cancer Cell Lines Reveals Episodic APOBEC Mutagenesis. *Cell* **176**, 1282-1294.e20 (2019).
 33. Alexandrov, L. B. *et al.* Signatures of mutational processes in human cancer. *Nature* **500**, 415–21 (2013).
 34. Kucab, J. E. *et al.* A Compendium of Mutational Signatures of Environmental Agents. *Cell* **177**, 821-836.e16 (2019).
 35. Chen, X. *et al.* Recurrent somatic structural variations contribute to tumorigenesis in pediatric osteosarcoma. *Cell Rep.* **7**, 104–112 (2014).

REVIEWER COMMENTS

Reviewer #2 (Remarks to the Author):

Thank you for asking me to review this paper one more time. I continue to question the novelty of this paper which merely confirms existing knowledge. I note that Reviewer 1 has the same fundamental concern.

The key issues are (lettering as per rebuttal document):

A) The authors continue to insist that their paper merits publication as a substantial resource. At a sample size of $n=702$ with the MAJORITY being exomes ($n=539$), this would not be considered an impressive resource in the context of childhood cancer genomics study. The authors are portraying only 205 genomes. Note that in 2012 (!) Molenaar et al had already reported 87 neuroblastoma genomes. A substantial meta-analysis would be a 2017 (!) study of ~500 medulloblastoma genomes (Northcott et al). Perhaps the authors may understand therefore that their study does not appear to be an impressive resource. Another issue with the WGS data of this study is that the WGS is mainly Complete Genomics (i.e. non-Illumina) data, which, as the authors know, is of limited use to the community.

C) The authors had stated that key novelty was: "We show for the first time the association of mitochondrial gene expression to mutational signature 18 in neuroblastoma, a significant finding on the potential mutagenic processes of neuroblastoma."

After the adjustments I had asked the authors to perform, this aspect of the analysis has become more questionable:

C.1) Association of MYCN/17q gain with signature 18 mutations: To me this association looks like it can be explained by "rearrangedness" (which would account for MYCN and 17q). Can the authors test whether the degree of "rearrangedness" (eg, number of breakpoints, degree of segmentation) would account for signature 18 mutations?

C.2) In testing the associations of various genomic features with signature 18, how many hypotheses have the authors tested? Is there multiple hypotheses correction?

C.3) As some of the "mitochondrial" genes that are associated with 17q gain reside on 17q, the authors need to show that the expression of 17q "mitochondrial" genes is enriched compared to "non-mitochondrial" 17q genes. Otherwise it would seem very questionable to claim that gains of 17q are associated with increased expression on 17q.

C.4) Looking at Figure 4C, it is not clear to me how the authors have identified "mitochondrial*" genes as significantly associated with signature 18 mutations. Where do the authors show that these are significantly enriched amongst the mRNAs shown in Figure 4C?

All other aspects of the paper reiterate existing knowledge, as previously detailed and graciously acknowledged by the authors.

RESPONSE TO REVIEWER COMMENTS

Reviewer #2 (Remarks to the Author):

Thank you for asking me to review this paper one more time. I continue to question the novelty of this paper which merely confirms existing knowledge. I note that Reviewer 1 has the same fundamental concern.

The key issues are (lettering as per rebuttal document):

A) The authors continue to insist that their paper merits publication as a substantial resource. At a sample size of $n=702$ with the MAJORITY being exomes ($n=539$), this would not be considered an impressive resource in the context of childhood cancer genomics study. The authors are portraying only 205 genomes. Note that in 2012 (!) Molenaar et al had already reported 87 neuroblastoma genomes. A substantial meta-analysis would be a 2017 (!) study of ~500 medulloblastoma genomes (Northcott et al). Perhaps the authors may understand therefore that their study does not appear to be an impressive resource. Another issue with the WGS data of this study is that the WGS is mainly Complete Genomics (i.e. non-Illumina) data, which, as the authors know, is of limited use to the community.

[Response]: We respectfully disagree that Complete Genomics (CGI) data “is of limited use to the community.” Our pan-pediatric landscape paper studying 1,699 pediatric cancers (Ma et al., *Nature* 2018) included 654 CGI-based WGS samples and has been cited 186 times in just over two years. In that paper, we documented extensive data cleaning techniques that make CGI data highly usable and allow effective detection of SNVs, indels, structural variants, and copy number alterations, which was incorporated into this study. We also note that 58 of the 205 WGS samples were obtained with Illumina sequencing. The reviewer also notes the Molenaar et al. study as an example; we would like to point out that this study also used CGI data for all 87 samples.

While the exome data do not enable identification of structural variants, they do allow sensitive and specific detection of large copy number alterations as documented in the Methods (59 samples had both WGS and exome data, enabling a comparison of the two platforms). The combined copy number, SNV/indel, and survival information (shown in Supplementary Table 1, with raw data also accessible on dbGaP and EGA) for this large cohort offer a valuable resource for the cancer research community.

C) The authors had stated that key novelty was: “We show for the first time the association of mitochondrial gene expression to mutational signature 18 in neuroblastoma, a significant finding on the potential mutagenic processes of neuroblastoma.”

After the adjustments I had asked the authors to perform, this aspect of the analysis has become more questionable:

C.1) Association of MYCN/17q gain with signature 18 mutations: To me this association looks like it can be explained by “rearrangedness” (which would account for MYCN and 17q). Can the authors test whether the degree of “rearrangedness” (eg, number of breakpoints, degree of segmentation) would account for signature 18 mutations?

[Response]: We have performed additional analysis limiting the signature 18 correlation analysis to stage 4 samples only. When doing this, 17q gains and *MYCN* alterations remain statistically associated with signature 18 (new Supplementary Fig. 18). Further, the structural variant burden was correlated very weakly with the signature 18 burden (Pearson $r^2 = 0.05$). We therefore conclude that the signature 18 correlations with 17q gain and *MYCN* are not related to the degree of rearrangements in neuroblastoma. We have included this information in the revised manuscript.

Location(s) of changes: Supplementary Fig. 18 and lines 337-340 (Results section “Mutational signature 18 is associated with increased mitochondrial gene expression, 17q gains, and *MYCN*”).

C.2) In testing the associations of various genomic features with signature 18, how many hypotheses have the authors tested? Is there multiple hypotheses correction?

[Response]: We tested the association of 17 genomic features with signature 18, including copy alterations such 17q gain, and gene-specific alterations such as *MYCN*, all of which are shown in Supplementary Fig. 17. We did not include any multiple hypothesis correction.

To address the reviewer’s concern, we have incorporated a Bonferroni correction for multiple hypothesis testing, thus requiring a P value of $\alpha = 0.05 / 17 = 2.94 \times 10^{-3}$ to consider a result statistically significant. Each of the alterations previously noted as being associated with signature 18 have P values below this more stringent threshold, as shown below, and hence no conclusions have changed.

- 17q gain $P = 1.3 \times 10^{-9}$
- *MYCN* $P = 4.4 \times 10^{-12}$
- 1p deletion $P = 5.6 \times 10^{-10}$
- 2p gain $P = 2.6 \times 10^{-4}$

We have noted the multiple hypothesis correction in the revised manuscript.

Location(s) of changes: Lines 335-337 (Results section “Mutational signature 18 is associated with increased mitochondrial gene expression, 17q gains, and *MYCN*”).

C.3) As some of the “mitochondrial” genes that are associated with 17q gain reside on 17q, the authors need to show that the expression of 17q “mitochondrial” genes is enriched compared to “non-mitochondrial” 17q genes. Otherwise it would seem very questionable to claim that gains of 17q are associated with increased expression on 17q.

[Response]: To address this concern, we divided 17q genes into two groups: those with mitochondrial localization based on the Broad Institute’s MitoCarta 2.0 database ($n = 29$ genes), and those without mitochondrial localization ($n = 344$ genes). We then analyzed the adjusted P values resulting from each gene’s differential expression between 17q gain samples vs. non-gained samples. (All 29 mitochondrially-localized genes had increased expression in 17q-gained samples, while 320 of 344 non-mitochondrial genes did.) The median P value for the mitochondrially-localized 17q genes was $P = 0.006$, while for non-mitochondrial 17q genes it was $P = 0.038$. A Wilcoxon rank-sum test comparing the P values between the two groups (a P value of P values) was $P = 0.042$.

Thus, mitochondrial 17q genes are more statistically increased upon 17q gain than non-mitochondrial genes. We have noted this in the revised manuscript.

Location(s) of changes: Lines 355-358 (Results section “Mutational signature 18 is associated with increased mitochondrial gene expression, 17q gains, and *MYCN*”).

C.4) Looking at Figure 4C, it is not clear to me how the authors have identified “mitochondrial*” genes as significantly associated with signature 18 mutations. Where do the authors show that these are significantly enriched amongst the mRNAs shown in Figure 4C?

[Response]: As noted in the manuscript, Figure 4C shows that genes in red (those statistically increased in signature 18-positive samples with adjusted $P < 0.003$ as shown on the graph) include several genes involved in mitochondrial function, including ATP synthase subunit *ATP5G1*; mitochondrial ribosome genes *MRPS7*, *MRPL11*, and *ICT1*; electron transport chain component *COX8A*; and *MTFP1*, a protein involved in mitochondrial fission.

We also note that 16.4% of all genes statistically increased in signature 18-positive samples with adjusted $P < 0.003$ have mitochondrial localization according to the Broad Institute's MitoCarta 2.0 database, compared to only 6.8% of all other genes ($P = 0.016$ by Fisher's exact test). We have included this information in the revised manuscript.

Location(s) of changes: Lines 319-323 (Results section "Mutational signature 18 is associated with increased mitochondrial gene expression, 17q gains, and *MYCN*").

All other aspects of the paper reiterate existing knowledge, as previously detailed and graciously acknowledged by the authors.

REVIEWERS' COMMENTS:

Reviewer #2 (Remarks to the Author):

I continue to question the merit of the paper which I don't think adds a great deal to the literature. At this juncture, what the authors need is an editorial decision.

RESPONSE TO REVIEWER

REVIEWERS' COMMENTS:

Reviewer #2 (Remarks to the Author):

I continue to question the merit of the paper which I don't think adds a great deal to the literature. At this juncture, what the authors need is an editorial decision.

[Response]: We appreciate reviewer #2's helpful feedback regarding technical issues, and are grateful that all technical issues are now resolved.